# Identification of Negative Transfers in Multitask Learning Using Surrogate Models

**Dongyue Li**                                                      *li.dongyu@northeastern.edu*
*Northeastern University, Boston*

**Huy L. Nguyen**                                                   *hu.nguyen@northeastern.edu*
*Northeastern University, Boston*

**Hongyang R. Zhang**                                               *ho.zhang@northeastern.edu*
*Northeastern University, Boston*

**Reviewed on OpenReview:** *https://openreview.net/forum?id=KgfFAI9f3E*

## Abstract

Multitask learning is widely used in practice to train a low-resource target task by augmenting it with multiple related source tasks. Yet, naively combining all the source tasks with a target task does not always improve the prediction performance for the target task due to negative transfers. Thus, a critical problem in multitask learning is identifying subsets of source tasks that would benefit the target task. This problem is computationally challenging since the number of subsets grows exponentially with the number of source tasks; efficient heuristics for subset selection does not always capture the relationship between task subsets and multitask learning performances. In this paper, we introduce an efficient procedure to address this problem via surrogate modeling. In surrogate modeling, we sample (random) subsets of source tasks and precompute their multitask learning performances; Then, we approximate the precomputed performances with a linear regression model that can also be used to predict the multitask performance of unseen task subsets. We show theoretically and empirically that fitting this model only requires sampling linearly many subsets in the number of source tasks. The fitted model provides a relevance score between each source task and the target task; We use the relevance scores to perform subset selection for multitask learning by thresholding. Through extensive experiments, we show that our approach predicts negative transfers from multiple source tasks to target tasks much more accurately than existing task affinity measures. Additionally, we demonstrate that for five weak supervision datasets, our approach consistently improves upon existing optimization methods for multi-task learning.

## 1 Introduction

Multitask learning (MTL) is an approach to combining several tasks together and learning one model for all tasks simultaneously (Caruana, 1997). The premise is that by combining the data samples of several tasks together, the dataset size of each task increases, thus improving the learning performance for every task. However, naively using all the source tasks may worsen performance compared to single-task learning (STL) for target tasks if there exist source tasks that are unrelated to them. This problem is commonly referred to as negative transfers in the literature but is challenging to predict for many tasks (Rosenstein et al., 2005).

The importance of developing a better understanding of multiple learning performance is well recognized. In the seminal work of Caruana (1997, Chapter 3), heuristics for judging "task-relatedness" in several applications are discussed; For instance, tasks that share many input features are more likely to be related to each other. A classical result by Ben-David et al. (2010) introduces an $\mathcal{H}$-divergence notion that quantifies the distance between two label distributions and relates the bias of source domains to this notion. MTL

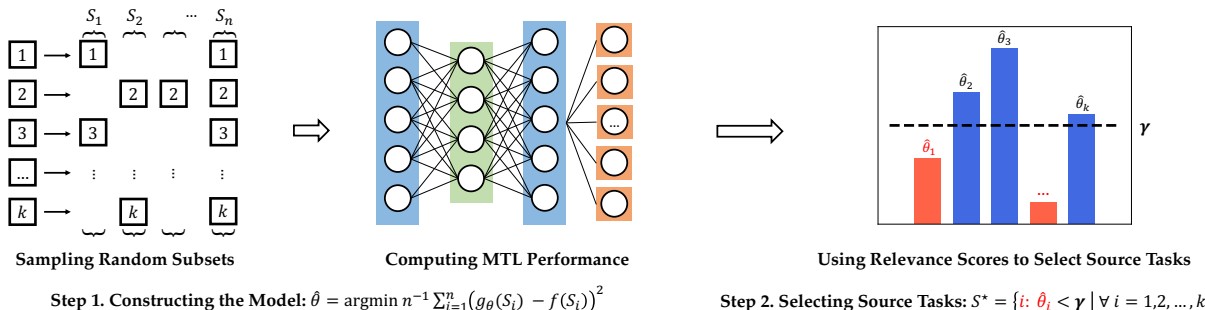

**Step 1. Constructing the Model:** $\hat{\theta} = \underset{\theta}{\mathrm{argmin}} \; n^{-1} \sum_{i=1}^{n} \big(g_\theta(S_i) - f(S_i)\big)^2$

**Step 2. Selecting Source Tasks:** $S^\star = \big\{i \colon \theta_i < \gamma \mid \forall\, i = 1, 2, \ldots, k\big\}$

Figure 1: Our approach involves two steps, as shown in the figure above. In the first step, we sample $n$ random subsets of source tasks among $k$ source tasks. Let these random subsets be denoted as $S_1, S_2, \ldots, S_n$. For $i$ from 1 up to $n$, we train an MTL model using the combined data of all the tasks in $S_i$ and the target task. We evaluate the loss of this MTL model on an evaluation set from the target task and denote the evaluation loss as $f(S_i)$. Then, we train a linear regression model to estimate the relevance scores $\theta_1, \theta_2, \ldots, \theta_k$, by minimizing the mean squared error between $g_\theta(S_i) = \sum_{j \in S_i} \theta_j$ and $f(S_i)$, as shown on the left. Let the estimated scores be denoted as $\hat{\theta} = [\hat{\theta}_1, \hat{\theta}_2, \ldots, \hat{\theta}_k]$. To predict the MTL performance of any unseen subset $S$, we compute $g_{\hat{\theta}}(S) = \sum_{i \in S} \hat{\theta}_i$. In the second step, we perform subset selection by choosing any task whose relevance score is below a threshold $\gamma$; We choose this selection criterion by examining the subset $S$ that minimizes $g_{\hat{\theta}}(S) = \sum_{j \in S} \hat{\theta}_j$, as shown in the right figure.

may perform worse than STL if the bias is too large. In weakly-supervised learning, several programmatic labeling functions are used to annotate a corpus of unlabeled data, and each labeling function can be treated as a source task (Ratner et al., 2016). The task labels can be highly noisy, causing negative transfer during training even though the tasks share the same input features (Ratner et al., 2019). In multitask learning of text prediction tasks, negative transfers are observed between different categories of tasks (e.g., question answering vs. sequence labeling) and different sizes of datasets (Vu et al., 2020).

Motivated by the need to reduce negative transfers between different tasks, researchers have developed optimization methods for multi-task learning from various fields. The most thorough approach to addressing these issues is to train all possible combinations of source tasks with the target task and find which subset of source tasks improves performance on the main target task. If there are $k$ source tasks, then this approach requires training $2^k$ MTL models. This is impractical (e.g., when $k \geq 20$). A more efficient solution is to train combinations of every *single* source task with the target task to determine if one task helps and then merge the helpful tasks together. This approach captures pairwise transfers, which measures first-order task affinities from one source task to another task (Fifty et al., 2021). Regarding higher-order transfers from multiple source tasks to another task, approximation techniques such as averaging the first-order affinity scores of each source task have been explored (Standley et al., 2020).

This paper designs and analyzes a scalable approach to identify negative transfers from multiple source tasks to one target task. The key idea is to construct a surrogate model to approximate the MTL performance of a subset of source tasks combined with the target task. Compared with prior works that measure task-relatedness based on either gradient similarity (Yu et al., 2020) or feature space alignment of neural networks (Nguyen et al., 2020; Raghu et al., 2020; Wu et al., 2020), our approach can be used to identify negative transfers from a set of tasks to another task. It also differs from existing discrepancy notions (e.g., $\mathcal{H}$-divergence) between source and target domains, which are difficult to measure for deep neural networks. Our approach builds on a recent paper that designs datamodels to predict the predictions of deep neural networks trained on a subset of training data (Ilyas et al., 2022). Unlike their work that focuses on single-task learning, we build surrogate models for multi-task learning with deep neural networks.

The first step of our approach involves learning a relevance score between every source task and the target task while accounting for the presence of the other source tasks. Let $\theta_i$ denote the relevance score of task $i$, for $i$ from 1 to $k$, where $k$ is the total number of source tasks. Conceptually, $\theta_i$ is analogous to the important

score of a feature in random forests when hundreds of other features are available. To estimate the relevance scores, we introduce a surrogate model $g_\theta(S)$, parametrized by the relevance scores $\theta$, to approximate MTL performances. Given any subset of source tasks $S$, let $f(S)$ be a loss function that measures the performance of combining $S$ and the target task to train an MTL model and then evaluated on the target task. The value of $f(S)$ provides a relevance measure between $S$ and the target task. Recall that $\theta_i$ measures the relevance of task $i$ to the target task. Thus, a lower value of $\theta_i$ indicates a higher relevance of task $i$ to the target task.

We specify a linear surrogate model as $g_\theta(S) = \sum_{j \in S} \theta_j$ (parametrized by the relevance scores) and minimize the mean squared error between $g_\theta(S_i)$ and $f(S_i)$ over $n$ random subsets, for $i$ from 1 to $n$. In particular, we precompute the values of $f(S_1), f(S_2), \ldots, f(S_n)$ by training one MTL model for each subset. We use such a linear specification of $\theta$ because computing the performance of each subset requires training an MTL model, which is not scalable unless $n$ grows almost linearly in $k$. In addition, we take inspiration from the recent work on datamodels (Ilyas et al., 2022), which shows that a linear regression model can extrapolate the predictions of deep neural networks for subsets of training data. We rigorously analyze the sample complexity of our approach in Theorem 2.1. After fitting $\theta$, we predict the performance of an unseen subset $S$ as $g_\theta(S)$ and compare it with the STL performance of the target task to determine if $S$ provides a negative transfer.

The second step of our approach involves selecting a subset of source tasks by choosing any source task whose relevance score is below a threshold $\gamma$. We derive this selection criterion by examining the minimum of the surrogate model $g_\theta(S)$ over all possible subsets $S$. We analyze this algorithm in a setting that includes one group of source tasks closer to the target task and another group further from the target task. The analysis reveals that for each task $i$, its relevance score $\theta_i$ is proportional to the sum of the MTL performances of all subsets that include $i$. Moreover, these performances preserve the distance gaps from the source tasks. See Theorem 3.1 for the precise result. In practice, we pick $\gamma$ via cross-validation; See Section 4 for the range of $\gamma$ that we validate on in the experiments. Taken together, our approach provides an efficient pipeline to predict and optimize multitask learning performances for task subsets. See Figure 1 for an illustration.

**Experimental Results.** We conduct extensive experiments to validate our approach in numerous data modalities and performance metrics. We summarize a list of our results as follows:

- The runtime for constructing surrogate models until convergence scales linearly in $k$, and the predicted performances accurately fit the true MTL performances of unseen subsets, measured by Spearman's correlation (0.8 averaged among 16 evaluations). Our approach achieves 4 times higher accuracy for predicting positive vs. negative transfers than known approximation schemes, measured by the $F_1$-score.

- By selecting source tasks based on the predicted MTL performances and only using the selected source tasks, we observe consistent benefits over existing optimization methods. We evaluate our approach on many datasets, including weak supervision, NLP, and multi-group fairness. In addition, we apply our approach to different MTL encoders, including BERT and multi-layer perceptrons. Notably, we consider a weak supervision dataset with as many as 164 labeling functions (Zhang et al., 2021). By selecting labeling functions with our approach and then applying MTL, we obtain up to 3.6% absolute accuracy lift compared with existing methods.

- We further visualize the tasks selected by our approach and find a separation between the selected tasks in terms of their labeling accuracies. Besides, our approach can also be used in scenarios where multiple groups of heterogeneous subpopulations are present. We are interested in the fairness and robustness of the learned model, measured as the performance of the worst-performing group. We apply our MTL framework as an augmentation to expand the dataset size of the worst-performing group and show consistent empirical performance in the worst-group accuracy metric.

**Summary of Contributions.** To summarize, this paper makes three contributions to studying negative transfers in multi-task learning. First, we study the higher-order task relationships from a set of source tasks to another task. We meta-learn such relationships using a linear regression method that can also predict an unseen subset's MTL performance. Second, we design a subset selection criterion for multi-task learning, which adjusts a threshold on the averaged MTL performances of subsets of tasks. Compared with the existing literature, our approach is better at predicting higher-order task relationships (See Figure 3 in Section 3 for the detailed result). Third, we validate our approach with extensive theoretical and experimental results.

**Organization.** We describe the problem setup and the surrogate model in Section 2. In Section 3, we use the relevance scores to perform subset selection for multitask learning. We present the experiments in Section 4. Then, we discuss the related works in Section 5. Lastly, we summarize the paper in Section 6. The appendix provides complete proof of our theoretical results and omitted results from the experiments.

## 2 Predicting Multitask Learning Performances Using Surrogate Models

This section describes the design and analysis of surrogate models for multitask learning. We begin by defining the problem setup. Then, we describe the construction of surrogate models and the estimation of the relevance scores. Lastly, we analyze the sample complexity of the construction procedure.

### 2.1 Preliminaries

**Problem Setup.** Let $t = 0$ denote the main target task of interest. Suppose the task's features and labels are drawn from an unknown distribution, denoted as $\mathcal{D}_t$. Let $\mathcal{X}$ denote the feature space. Let the set of all possible labels be denoted as $\mathcal{Y}$. We are given a dataset, which includes a list of examples drawn independently from $\mathcal{D}_t$. Besides, we are also given $k$ datasets from related source tasks, which are all supported on $\mathcal{X} \times \mathcal{Y}$.

A naive approach to optimize MTL is combining all the datasets and evaluating the trained model on the target task. However, this might result in worse performance than single-task learning. Thus, it is crucial to identify if a source task would help or hurt. The most thorough solution for addressing this question is by enumerating all possible combinations of source tasks, leading to a total of $2^k$ combinations. For each combination of source tasks, train a multitask model using the selected source tasks and the main task. While this procedure optimizes the performance of MTL, it is too slow for large $k$.

How can we optimize the performance of MTL efficiently? Relatedly, given a set of source tasks, can we predict their transfer effects upon the target task efficiently? Below, we define two common transfer effects.

**Positive vs. Negative Transfer.** Consider any multitask learning algorithm, denoted as $\mathcal{A}$, which trains a joint model given any set of tasks. For any subset $S \subseteq \{1, 2, \ldots, k\}$, we say that $S$ provides a negative transfer to $t$ if the performance of $\mathcal{A}(S \cup \{t\})$ is worse than $\mathcal{A}(\{t\})$ (e.g., in terms of higher loss values). Likewise, we say that $S$ provides a positive transfer to $t$ if the performance of $\mathcal{A}(S \cup \{t\})$ is better than $\mathcal{A}(\{t\})$. We aim to design a scalable method to predict such positive and negative transfer effects.

It is worth highlighting that both types of transfers are often observed in practice. To give an example, we consider a binary classification dataset that involves a total of 51 tasks. We pick one of them as the target task, use the rest as source tasks, and consider the case where $|S| = 1$. This leads to training 50 models for each target task, one for every combination of one source task and the target task. The results are shown in Figure 2, which provides illustrations for four different target tasks. The $y$-axis corresponds to the accuracy difference between the MTL and STL results. We consistently find a mix of positive and negative transfers for all four target tasks.

**Surrogate Models.** A recent paper by Ilyas et al. (2022) designs a linear regression method to predict the predictions of deep neural networks trained on a subset of training data. Surprisingly, an empirical finding from that paper is that this linear regression model provides a very good fit on a number of popular benchmark datasets such as CIFAR. This finding is later explored by another recent paper (Saunshi et al., 2022). We defer further discussions to the related works in Section 5. Both of these works focus on single-task supervised learning. Next, we will apply this idea of linear surrogate models to multi-task learning.

### 2.2 Constructing the Linear Surrogate Model

First, we construct a surrogate model to fit multitask learning performances. Let $\phi$ be an encoder that is shared by all tasks. For any input features $x \in \mathcal{X}$, the encoder $\phi$ maps $x$ into a feature vector. For every source task in $S$ and the target task, there is a separate prediction layer for each of them. Let $\psi_0, \psi_1, \psi_2, \ldots, \psi_k$ denote the prediction layers, which map the feature vectors to the output.

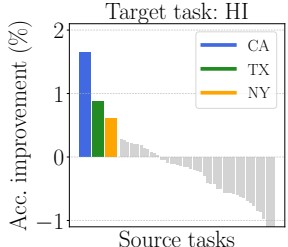 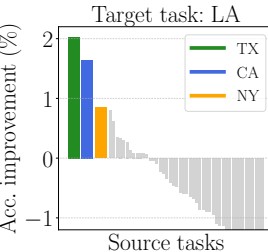 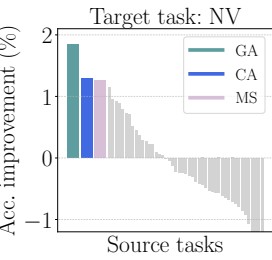 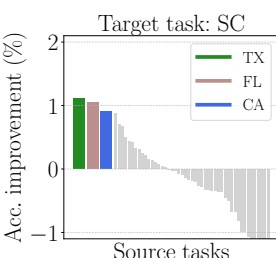

Figure 2: Illustration of mixed outcomes in multitask learning: In each figure, we train 50 multitask models based on one fixed target task and one source task from 50 source tasks. **$x$ axis**: Each bar represents one source task from 50 source tasks. **$y$ axis**: Test accuracy of MTL with one source task minus the test accuracy of STL with the target task alone. For further description of the experiment setup, see Section 4.1.

Given any subset $S$, we train $\phi$ and $\psi_i$, for $i \in S \cup \{0\}$, by minimizing the average loss over the combined training data along with the target task. Let $\phi^{(S)}$ and $\psi_i^{(S)}$, for $i \in S \cup \{0\}$, be the trained model. We evaluate its loss on the target task's validation dataset. Let $\widetilde{\mathcal{D}}_t = \{(x_1, y_1), \ldots, (x_m, y_m)\}$ denote a set of $m$ independent samples from $\mathcal{D}_t$, which is used as a validation set for the target task. Let $\ell$ be a non-negative loss such as the cross-entropy loss. We define *multitask learning performances* as:

$$f(S) = \frac{1}{m} \sum_{i=1}^{m} \ell\Big(\psi_0^{(S)}\big(\phi^{(S)}(x_i)\big), y_i\Big), \text{ for any } S \subseteq \{1, 2, \ldots, k\}. \tag{1}$$

Our main idea is to construct a surrogate model, parametrized by a relevance score $\theta_i$ for each source task $i$. We use a linear specification inspired by recent work of Ilyas et al. (2022)[1]

$$g_\theta(S) = \sum_{i \in S} \theta_i, \text{ where } \theta_i \text{ is the } i\text{-th entry of } \theta, \text{ for any } i = 1, 2, \ldots, k. \tag{2}$$

The procedure for estimating $\theta$ is as follows. First, sample $n$ subsets of source tasks from $\{1, 2, \ldots, k\}$, denoted as $S_1, S_2, \ldots, S_n$. We sample each subset from the uniform distribution over subsets with a fixed size of $\alpha$; We will justify this choice later in Section 3.2. For instance, to capture the transfer from five source tasks to the target task, we can set $\alpha = 5$. Then, compute the value of $f(S_i)$ by training one MTL model for every $i = 1, 2, \ldots, n$. Lastly, minimize the mean squared error (MSE) between $g_\theta(S_i)$ and $f(S_i)$, averaged over all $i$:

$$\hat{\mathcal{L}}_n(\theta) = \frac{1}{n} \sum_{i=1}^{n} \Big(g_\theta(S_i) - f(S_i)\Big)^2. \tag{3}$$

Let $\hat{\theta}$ denote the minimizer of the above MSE. For brevity, we refer to $\hat{\theta}$ as the task model. After estimating $\hat{\theta}$, for an unseen subset of source tasks $S$, we predict its MTL performance as $g_{\hat{\theta}}(S) = \sum_{i \in S} \hat{\theta}_i$.

## 2.3 Sample Complexity

We show that given $n = \mathrm{O}\big(k \log^2(k)\big)$, we can estimate $\hat{\theta}$ accurately. To be precise, let $\mathcal{U}$ denote the uniform distribution over all subsets of size $\alpha$ drawn from $\{1, 2, \ldots, k\}$. Let $T$ denote an unseen subset drawn from $\mathcal{U}$. The population risk for a given $\theta$ is defined as the expected MSE between $f(T)$ and $g_\theta(T)$:

$$\mathcal{L}(\theta) = \mathbb{E}_f \mathbb{E}_T \Big[\big(f(T) - g_\theta(T)\big)^2\Big]. \tag{4}$$

Let the minimizer of the above risk be denoted as $\theta^\star$. We prove that $\hat{\theta}$ converges to $\theta^\star$ using Rademacher complexity-based arguments. Let the function class of $\psi_t$ and $\phi$ be denoted as $\mathcal{H}$. Let the loss function

---

[1]We use this specification for scalability consideration. Note that it is possible to consider more complex specifications, such as adding quadratic variables $\theta_{1,2}, \theta_{1,3} \ldots, \theta_{k-1,k}$. The construction procedure and the analysis is conceptually the same. However, the sample complexity for fitting these quadratic variables is $\mathrm{O}(k^2)$, rendering it infeasible for large $k$, e.g., $k \geq 100$.

class be $\mathcal{F} = \{\ell(\psi_t(\phi(x)), y) \mid \forall\, \psi_t, \phi \text{ from } \mathcal{H}\}$. Recall that $\tilde{\mathcal{D}}_t = \{(x_1, y_1), (x_2, y_2), \ldots, (x_m, y_m)\}$ refers to the dataset used to evaluate the value of $f(T)$, and its size is equal to $m$. Let $\sigma_1, \sigma_2, \ldots, \sigma_m$ be $m$ independent Rademacher random variables, collectively as $\sigma_{1:m}$. The Rademacher complexity of $\mathcal{F}$ over $\tilde{\mathcal{D}}_t$ is defined as

$$\mathcal{R}_m(\mathcal{F}) = \mathop{\mathbb{E}}_{\tilde{\mathcal{D}}_t} \mathop{\mathbb{E}}_{\sigma_{1:m}} \left[ \sup_{h \in \mathcal{F}} \frac{1}{m} \sum_{i=1}^m \sigma_i \cdot h(x_i, y_i) \right], \tag{5}$$

where the expectation is taken over the randomness of the empirically-drawn dataset $\tilde{\mathcal{D}}_t$ and the Rademacher random variables $\sigma_{1:m}$. We follow the convention of big-O notations for stating the result. Given two functions $h(n)$ and $h'(n)$, we use $h(n) = \mathrm{O}(h'(n))$ or $h(n) \lesssim h'(n)$ to indicate that $h(n) \leq C \cdot h'(n)$ for some fixed constant $C$ when $n$ is large enough. Our result is stated formally below.

**Theorem 2.1.** *Suppose the functions in $\mathcal{F}$ are all bounded from above by a fixed constant $C > 0$. Suppose $\alpha$ is less than $k/2$. Let $n$ be the number of sampled subsets and $m$ be the size of the set used to evaluate $f$. With probability at least $0.99$, $\hat{\theta}_n$ converges to $\theta^\star$ as $n, m$ are both large enough:*

$$\left\| \hat{\theta} - \theta^\star \right\| \lesssim C\sqrt{\frac{(k \log^2(k))\alpha^4}{n}} + \sqrt{\frac{(\log(k))\alpha}{m}} + \mathcal{R}_m(\mathcal{F}), \tag{6}$$

*where $\|\cdot\|$ denotes the Euclidean norm of a vector.*

Based on the above result, it is clear from equation (6) that provided with $\mathrm{O}(k \log^2(k))$ random samples, the first error term relating to $n$ shrinks to a negligible value (one may think of $\alpha$ as a fixed constant such as 5 or 10). There are two error terms decreasing with $m$, the size of $\tilde{\mathcal{D}}_t$ used to evaluate $f$. The Rademacher complexity $\mathcal{R}_m(\mathcal{F})$ is known to be of order $\mathrm{O}(m^{-1/2})$ when $\mathcal{H}$ represents a family of neural networks (Bartlett et al., 2017). These two error terms are due to the variance of $f$ since it is measured on a finite set. Lastly, we note that the probability value of 0.99 in the above theorem statement can be adjusted to other values. In the proofs, we state the result more generally for any probability value $1 - \delta$, where $\delta > 0$; See the statements of Lemma 2.2 and Lemma 2.3 below for details.

**Proof Overview.** We introduce a few notations to examine $g_\theta(S)$ and the covariance of $\mathcal{U}$. Let $\mathcal{I}_n \in \{0, 1\}^{n \times k}$ be a zero-one matrix; For any $i = 1, 2, \ldots, n$, the $i$-th row is $\mathbb{1}_{S_i}$, the characteristic vector of $S_i$. Let $\hat{f}$ be a vector in which $\hat{f}_i = f(S_i)$, for any $i = 1, 2, \ldots, n$. The $\hat{\theta}$ that minimizes equation (3) is equal to

$$\hat{\theta} = \left( \mathcal{I}_n^\top \mathcal{I}_n \right)^{-1} \mathcal{I}_n^\top \hat{f}. \tag{7}$$

Let $v = \mathcal{I}_n^\top \hat{f}$ and let $v_i$ be the $i$-th entry of $v$, for $i = 1, \ldots, k$. Based on the definition of $\mathcal{I}_n$, we observe that

$$v_i = \sum_{1 \leq j \leq n:\, i \in S_j} f(S_j), \quad \text{for any } 1 \leq i \leq k. \tag{8}$$

Next, let $\mathcal{I} \in \{0, 1\}^{|\mathcal{U}| \times k}$ be a zero-one matrix, where $|\mathcal{U}|$ is the number of subsets in $\mathcal{U}$. Each row of $\mathcal{I}$ corresponds to the characteristic vector of a subset. Let $f$ be a vector such that each entry of this vector corresponds to the MTL performances (cf. equation (1)) of a subset in distribution $\mathcal{U}$. The population risk minimizer $\theta^\star$ for reducing $\mathcal{L}(\theta)$ in equation (4) is equal to

$$\theta^\star = \left( \mathcal{I}^\top \mathcal{I} \right)^{-1} \mathcal{I}^\top \, \mathbb{E}[f].$$

Our proof involves two steps. First, we deal with the error due to the randomness of the random subsets. Let

$$\bar{\theta} = \left( \mathcal{I}^\top \mathcal{I} \right)^{-1} \mathcal{I}^\top f.$$

We state the following result, which shows that $\hat{\theta}$ converges to $\bar{\theta}$ as $n$ increases.

**Lemma 2.2.** *In the setting of Theorem 2.1, conditional on $f(T)$ for any subset $T \in \mathcal{U}$, with probability $1 - 2\delta$ over the randomness of $S_1, S_2, \ldots, S_n$, for any $\delta \geq 0$, the Euclidean distance between $\hat{\theta}$ and $\bar{\theta}$ satisfies:*

$$\left\|\hat{\theta} - \bar{\theta}\right\| \leq 4C\sqrt{\frac{(k\log^2(2k\delta^{-1}))\alpha^4}{n}} + 8C\sqrt{\frac{k\alpha^2}{\delta n}}. \tag{9}$$

The proof of the above result relies on a novel union bound taken over all subsets in $\mathcal{S}$. Crucially, there are at most $2^k$ subsets. By taking the logarithm of $2^k$ after the union bound, we get a factor of $k$ as shown in equation (9). Second, we prove the convergence from $\bar{\theta}$ to $\theta^\star$, as $m$ increases.

**Lemma 2.3.** *In the setting of Theorem 2.1, for any $\delta > 0$, with probability at least $1 - \delta$ over the randomness of $S_1, S_2, \ldots, S_n$ and $f(S_1), f(S_2), \ldots, f(S_n)$, the Euclidean distance between $\bar{\theta}$ and $\theta^\star$ satisfies:*

$$\left\|\bar{\theta} - \theta^\star\right\| \leq \frac{\mathcal{R}_m(\mathcal{F})}{\sqrt{2}} + \sqrt{\frac{\left(\log\left(\delta^{-1}k\right)\right)\alpha}{4m}}. \tag{10}$$

Combining Lemma 2.2 and Lemma 2.3 together, we have thus proved that equation (6) holds. The proof of the above two results can be found in Appendix B. This result justifies why we use a linear specification, as we can scale up the sample complexity.

**Remark 2.4.** The proof of Theorem 2.1 uses the design of the $\alpha$-sized subsets. In particular, the covariates of these $\alpha$-sized subsets are zero-one vectors, with the ones being drawn randomly. We show that the population covariance of all the $\alpha$-sized subsets is an identity matrix plus a rank-one matrix. See equation (12) in Section 3 for the derivation. This implies that the inverse of the covariance matrix is an identity matrix plus a rank-one matrix.

# 3 Subset Selection for Multitask Learning

Next, we design an algorithm to optimize the performance of the target task using the relevance scores of each source task. We analyze this algorithm in a setting where the tasks are separated into two groups, with one group being more similar (measured by Euclidean distances) to the target task than the other group.

## 3.1 Using Relevance Scores to Select Source Tasks

Provided with the surrogate model, we can optimize the target task performance by using the predicted MTL performances. A natural way is to select the subset that minimizes the value of $g_{\hat{\theta}}(S)$ over subset $S$. Due to the linear specification of $g_{\hat{\theta}}$, this is equivalent to selecting source tasks with a small $\hat{\theta}_i$. Thus, we select a source task $i$ if $\hat{\theta}_i$ is below the desired threshold $\gamma$, which can be determined via cross-validation. Then, we train a model by combining the selected source tasks with the target task. The complete procedure is shown below. We will rigorously justify the existence of a threshold afterward.

---

**Algorithm 1** Subset Selection for Multi-Task Learning Using Relevance Scores

---

**Input**: $k$ source tasks; Training and validation datasets of the target task.
**Require:** Size of each subset $\alpha$; Number of sampled subsets $n$; MTL algorithm $f$; Task selection threshold $\gamma$.
**Output**: Trained model $\phi^{(S^\star)}, \psi_t^{(S^\star)}$.
  1: For $i = 1, \ldots, n$, sample a random subset $S_i$ from $\{1, 2, \ldots, k\}$ with size $\alpha$; evaluate $f(S_i)$ following equation (1).
  2: Estimate the relevance scores $\hat{\theta}$ following equation (3).
  3: Select source tasks based on their relevance scores: $S^\star = \left\{i : \hat{\theta}_i < \gamma \mid \forall i = 1, 2, \ldots, k\right\}$.
  4: Train a model by combining $S^\star$ and $t$; denote the trained model as $\phi^{(S^\star)}$, and $\psi_i^{(S^\star)}$ for all $i \in S^\star \cup \{t\}$.

---

### 3.2 Analysis of the Algorithm

We analyze the above procedure in a synthetic setting where the data is generated from a linear process. We assume that the input features for each task are drawn from an isotropic Gaussian distribution with $p$ dimensions. For each task $i$ from 0 to $k$, let $\beta^{(i)} \in \mathbb{R}^p$ denote the linear process for task $i$. Given a data point from task $i$ with feature vector $x$, its label is generated as $y = x^\top \beta^{(i)} + \epsilon$, where $\epsilon$ is a random variable with mean 0 and variance $\sigma^2$. Suppose there are two groups of tasks depending on their distances to $\beta^{(t)}$, given by $a, b$ such that $b > a > 0$. For every $i = 1, \ldots, k$, task $i$ is called a *good* task if $\|\beta^{(i)} - \beta^{(t)}\| \le a$; On the other hand, $i$ is a *bad* task if $\|\beta^{(i)} - \beta^{(t)}\| \ge b$. We show that there exists a threshold that separates good tasks from bad tasks, stated formally as follows.

**Theorem 3.1.** *In the setting described above, suppose $f$ is bounded from above by a fixed constant $C > 0$. Suppose there are $d \gtrsim k \log k + p + a^4 k^4 (a^2 - b^2)^{-2}$ data points from every source and target task. Suppose $n \gtrsim C^2 k^2 (a^2 - b^2)^{-2}$ and $m \gtrsim p \log p$. With probability at least $0.99$, there exists a threshold $\gamma$ such that*

- *For any $i = 1, 2, \ldots, k$, if task $i$ is a good task, then $\hat{\theta}_i < \gamma$.*
- *Otherwise, if task $i$ is a bad task, then $\hat{\theta}_i > \gamma$.*

The intuition behind the above result is that $\hat{\theta}_i$ averages the MTL performances of all subsets involving $i$. If $i$ is a good task, the average performance will be lower, leading to a lower relevance score. Moreover, there exists a threshold that separates the relevance scores of good tasks and bad tasks. We give a toy example to illustrate why $\hat{\theta}$ can preserve the Euclidean distance gaps from $\beta$. Our experiments later also confirm the existence of such a separation (cf. Figure 5).

**Example 3.2** (One-dimensional case). *Consider a one-dimensional case where $p = 1$ and every $\beta$ is a real value. Let $\beta^{(t)} = 0$. Let $0 < \beta^{(i)} < a$ if $i$ is a good task. Let $b < \beta^{(i)}$ if $i$ is a bad task.*

- Our first observation is that $\hat{\theta}_i$ is proportional to $v_i$ (cf. equation (8)), as shown in Lemma 3.3.
- Our second observation is that $v_i$ is proportional to $\beta^{(i)}$. This is because $v_i$ is the sum of $f(S)$ among all $S \in \mathcal{U}$ involving $i$, and $f(S)$ is the average of $\beta^{(j)}$ among $j \in S$. Thus, $v_i$ is the average of all $\beta$'s from the $n$ random subsets, while $\beta^{(i)}$ has a larger weight in $v_i$ than the other $\beta$'s because $i$ is always in $S$.

Taken together, we conclude that the relevance scores can preserve the relative values of $\beta$ in this example.

**Proof Overview.** We now generalize the intuition from the one-dimensional case, beginning with the first observation. We show that the relevance scores preserve the distance gap of every pair of tasks from $\beta$.

**Lemma 3.3.** *In the setting of Theorem 3.1, with probability $1 - \delta$, for any $\delta > 0$, the following holds:*

$$\left| \frac{1}{n}(\hat{\theta}_i - \hat{\theta}_j) - \frac{k}{\alpha n}(v_i - v_j) \right| \lesssim \frac{\log(\delta^{-1} k)}{\sqrt{n}}, \text{ for any } 1 \le i < j \le k. \tag{11}$$

The above result analyzes the covariance of $\mathcal{I}_n$, which is proportional to identity plus a constant shift. Let $\mathrm{Id}_{k \times k}$ be a $k$ by $k$ identity matrix and $e \in \mathbb{R}^k$ be a vector whose entries are all equal to one. By the definition of $\mathcal{I}_n$ and Woodbury matrix identity, we have

$$\mathbb{E}\left[ \frac{\mathcal{I}_n^\top \mathcal{I}_n}{n} \right] = \frac{\alpha}{k} \mathrm{Id}_{k \times k} + \frac{\alpha(\alpha-1)}{k(k-1)} ee^\top \quad \Rightarrow \quad \mathbb{E}\left[ \frac{\mathcal{I}_n^\top \mathcal{I}_n}{n} \right]^{-1} = \frac{k}{\alpha}\left( \mathrm{Id}_{k \times k} - \frac{\alpha-1}{k\alpha-1} ee^\top \right). \tag{12}$$

Crucially, if we multiply $v$ on the right-hand side of equation (12), then we will get $\frac{k}{\alpha}(v - \frac{\alpha-1}{k\alpha-1}(e^\top v)e)$. Recall that $e$ is all one's vector, which has the same entry in every coordinate after rescaling. Then, recall $\hat{\theta}$ from equation (3). By matrix concentration inequalities, the spectral norm of the deviation from $\frac{\mathcal{I}_n^\top \mathcal{I}_n}{n}$ to its expectation satisfies

$$\left\| \frac{\mathcal{I}_n^\top \mathcal{I}_n}{n} - \mathbb{E}\left[ \frac{\mathcal{I}_n^\top \mathcal{I}_n}{n} \right] \right\|_2 \lesssim \frac{\alpha \log(k\delta^{-1})}{\sqrt{n}}. \tag{13}$$

See equation (20), Appendix B.1 for the proof. Thus, combining equations (12) and (13), we claim that $\hat{\theta}$ is equal to $\alpha^{-1}kv$ minus a shared term for every task, modulo the deviation error of order $\mathrm{O}(n^{-1/2})$. By subtracting $\hat{\theta}_i - \frac{k}{\alpha}v_i$ and $\hat{\theta}_j - \frac{k}{\alpha}v_j$, we can cancel out the shared term, leading to equation (11).

Next, we formalize the second observation from the one-dimensional case. Based on equation (8), $v_i$ is a sum of $f(S)$ for all subsets $S$ such that $i \in S$. We then show that $f(S)$ is the sum of $\beta^{(j)}$ for all $j \in S$, based on the pooling structure of our MTL model. Thus, the Euclidean distance between $\beta^{(i)}$ and $\beta^{(t)}$ will also reflect in $v_i$. For complete proof of Theorem 3.1 (and Lemma 3.3), see Appendix B.4. This result substantiates our intuition that $\hat{\theta}_i$ provides the relevance score of each source task $i$ to the target task while accounting for the presence of other source tasks.

**Remark 3.4.** After identifying the related tasks from all the source tasks, we can then combine them together with the target task for multi-task learning. Given that the distance between their $\beta$-coefficients is small enough (i.e., $a$ is small enough), we can show that MTL performs better than single-task learning.

## 4 Experiments

We apply our approach to three settings. The first setting is about applying weak supervision to unlabeled data, and we apply our algorithm to select labeling functions for combining the weak labels of the labeling functions. The second setting involves language prediction tasks from NLP benchmarks. Again, we use our algorithm to select source tasks to improve the performance of target tasks. The third setting involves learning from multiple groups of heterogeneous subpopulations, where the goal is to train a model with robust performance across all groups. We cast this multi-group learning problem into an MTL framework and apply our algorithm to select a subset of groups to improve the robustness of target tasks. For all these settings, we show that surrogate models can predict negative transfers accurately and fit MTL performances well; Moreover, our approach provides consistent benefits over various optimization methods for multi-task learning. The code repository for reproducing our experiments can be found at https://github.com/NEU-StatsML-Research/Task-Modeling.

### 4.1 Experimental Setup

**Datasets.** First, we apply our approach to several text classification tasks from a weak supervision dataset (Zhang et al., 2021). Each dataset uses several labeling functions to create labels for every unlabeled example. The labels generated by different labeling functions may conflict with each other. We view each labeling function as a source task. The goal is to predict an unlabeled set of examples which is viewed as the target task. A validation dataset that includes the correct labels is available for cross-validation. We include the dataset statistics in Table 1.

Second, we consider MTL with natural language processing tasks. We collect twenty-five datasets across a broad range of tasks, spanning sentiment classification, natural language inference, question answering, etc., from GLUE, SuperGLUE, TweetEval, and ANLI. We view one task as the target and the rest as source tasks. The goal is to select a subset of source tasks for the best MTL performance. We provide the statistics of the twenty-five tasks in Table 4, Appendix C.1.

Third, we consider multi-group learning settings in which a dataset involves multiple subpopulation groups. We consider income prediction tasks based on US census data (Ding et al., 2021). The goal is to predict whether an individual's income is above \$50,000 using ten features, including the individual's education level, age, sex, etc. There are 51 states in this dataset; we view each state as one task. For prediction, we use one state as the target task and the remaining fifty states as source tasks. We use the racial group of each individual to split a state population into nine subpopulation groups. We evaluate the robustness of a model by the worst-group accuracy. This metric measures the accuracy of the worst-performing group among all groups. We use six states as the target task. See Table 2 for dataset statistics.

**Implementation.** We use a standard approach for conducting MTL, i.e., hard parameter sharing. For text classification, we use BERT-Base as the encoder. For tabular features, we use a fully-connected layer with a hidden size of 32. The surrogate modeling procedure requires three parameters: the size of a subset,

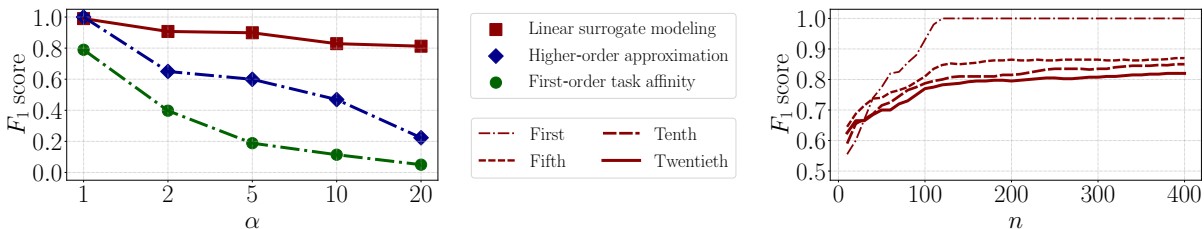

Figure 3: **Left**: Our approach can consistently predict positive/negative transfers from up to 20 source tasks to the target task. **Right**: Convergence of surrogate models as $n$ increases up to 400, leading to an $F_1$-score of 0.8 for predicting positive/negative transfers from up to 20 source tasks to one target task.

the number of samples, and the loss function. We select the size from a range between $3, 5, 10$, and $15$. We select the number of samples from a range between $50, 200, 400$, and $800$, depending on $k$. We also collect a holdout set of size 100 for constructing the surrogate model. For classification tasks, we set the loss function as the negative classification margin, i.e., the difference between the correct-class probability and the highest incorrect-class probability. After estimating the surrogate model $g$ from equation (3), we use $g(S)$ as the predicted multitask loss for an unseen subset $S$. We compare $g(S)$ with the STL performance of task $t$ to determine whether the transfer from $S$ to $t$ is positive or negative. We measure the $F_1$-score for the minority class (between the positive and negative classes) on the holdout set.

### 4.2 Results for Predicting Negative Transfers

We validate that our fitted models can accurately identify positive vs. negative transfers from source tasks. Then, we show that these models can be constructed efficiently by reporting the runtime.

**Results.** We test the accuracy of using surrogate models to predict positive vs. negative transfers. We first evaluate the four examples shown in Figure 2. We set the size of $\alpha$ as 5 and $n$ as 400. Using the model to compare the MTL performances with STL performances, we can correctly predict the transfers with an $F_1$-score of **0.82**, averaged over the four target tasks. Second, we conduct the same tests for weak supervision and NLP tasks. Similarly, we find that task models can predict positive vs. negative transfers with $F_1$-score of **0.8** on average for ten different target tasks.

Furthermore, we compare these results with two baselines that either compute first-order task affinity scores or higher-order approximations by averaging first-order affinity scores. Our approach yields much more accurate predictions across different subset sizes of $\alpha$, ranging from 5 up to 20. Figure 3 provides the illustration for one target task, which is conducted on the US Census dataset, along with fifty source tasks.

Lastly, we measure Spearman's correlation between the predicted performances and true performances. We observe an average coefficient of **0.8** across 16 target tasks. See Appendix A for the details.

**Computational Cost.** Next, we report the runtime cost collected on an NVIDIA Titan RTX card. First, we show that the running time of our procedure scales linearly with $k$, the number of source tasks. Recall that our approach requires training $n$ models, one for each random subset. We find that collecting $n \leq 8k$ samples suffice for the fitted model to converge. The results hold for 16 target tasks shown in Figure 6 of Appendix A. We plot the number of hours w.r.t. $k$ in Figure 4. The results confirm our hypothesis.

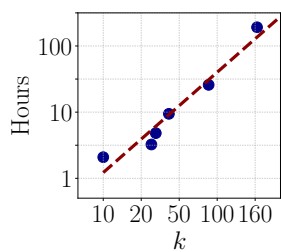

Our approach is also comparable with the baseline approaches. Among them, the most related ones compute first-order affinity scores and conduct a branch-and-bound search algorithm over the task space, which has exponential complexity in $k$ (Standley et al., 2020; Fifty et al., 2021). In our experience, with more than 20 tasks, these methods take more than 200 hours. Our approach requires at most 145 hours. This is consistent with our theoretical predictions in Section 2. Later in Section 4.5, we elaborate on ways to speed up the training.

Figure 4: We show that the runtime of our approaches scales linearly with $k$, the number of source tasks.

Table 1: Accuracy/F1-score from surrogate modeling followed by task selection (ours), as compared with MTL methods and weak supervision methods that use a label model to aggregate the weak labels.

| Dataset (Metrics) | Youtube (Acc.) | TREC (Acc.) | CDR (F1) | Chemprot (Acc.) | Semeval (Acc.) |
|---|---|---|---|---|---|
| Training | 1,586 | 4,965 | 8,430 | 12,861 | 1,749 |
| Validation | 120 | 500 | 920 | 1,607 | 178 |
| Test | 250 | 500 | 4,673 | 1,607 | 600 |
| # source tasks | 10 | 68 | 33 | 26 | 164 |
| Naive MTL | 94.72±0.85 | 64.10±0.50 | 58.20±0.55 | 53.43±0.53 | 89.00±1.06 |
| HOA | 94.93±1.80 | 74.67±4.66 | 59.76±0.97 | 45.57±0.41 | 89.94±4.42 |
| TAG | 95.20±0.65 | 77.50±3.62 | 59.31±0.15 | 53.67±2.74 | 89.06±1.47 |
| TAWT | 94.53±1.05 | 72.40±2.36 | 59.85±0.30 | 53.76±2.96 | 86.83±1.78 |
| Auto-$\lambda$ | 95.80±0.85 | 73.70±0.67 | 59.07±0.05 | 52.50±1.28 | 87.91±0.66 |
| Majority voting | 95.36±1.71 | 66.56±2.31 | 58.89±0.50 | 57.32±0.98 | 85.03±0.83 |
| Probabilistic modeling | 93.84±1.61 | 68.64±3.57 | 58.48±0.73 | 57.00±1.20 | 83.93±0.83 |
| MeTaL | 92.32±1.44 | 58.28±1.95 | 58.48±0.90 | 56.17±0.66 | 71.74±0.57 |
| **Alg. 1 (Ours)** | **97.47±0.82** | **81.80±1.14** | **61.22±0.39** | **57.54±0.55** | **93.50±0.24** |

### 4.3 Results for Improving MTL Performance

Next, we apply our approach to MTL on weak supervision and NLP tasks. We compare our approach with the following baselines. First, we consider training by naively combining all source and target tasks. Second, we consider bilevel optimization methods, including TAWT (Chen et al., 2022) and Auto-$\lambda$ (Liu et al., 2022), and MTL optimization methods, including HOA (Standley et al., 2020), TAG (Fifty et al., 2021). The latter two methods use a branch-and-bound algorithm that does not scale to over 20 tasks in one dataset. To allow for a comparison with them, we apply the thresholding procedure to their first-order task affinity scores to select source tasks. To set the threshold $\gamma$ in our algorithm, we use grid search from $-0.5$ to $0.5$ at an interval of 0.1. We choose this range because it covers the values of most coefficients in our experiments.

**Multitask weak supervision.** First, we apply our algorithm to five weak supervision datasets, which involve text classification from multiple weak labels. We select a subset of labeling functions so that using their weak labels to train an end model best improves performance on the target task. We also compare against methods that use a label model to aggregate the weak labels and then train an end model on the aggregated label. These include taking a majority vote on the weak labels, applying probabilistic modeling to combine the noisy labels (Ratner et al., 2016), and MeTaL (Ratner et al., 2019).

Next, we compare the experimental results, shown in Table 1. Compared with naively MTL, which trains all tasks together, our algorithm improves the test performance by **6.4%** on average. Compared with MTL optimization and weak supervision methods, our algorithm outperforms their results by up to **3.6%** absolute and **2.3%** on average.

Lastly, we examine the labeling functions selected by our approach. Recall that our procedure places a threshold over the coefficients to separate good and bad source tasks. Here, we use the number of correct and incorrect labels as a proxy of relatedness between a labeling function and the target task. Figure 5 shows the results, measured on two datasets, namely Chemprot and TREC. Each dot represents one source task. We observe a clear separation between selected and excluded source tasks when we compare the correct/incorrect labels in each task. This shows that our algorithm selects more accurate labeling functions for multi-task weak supervision.

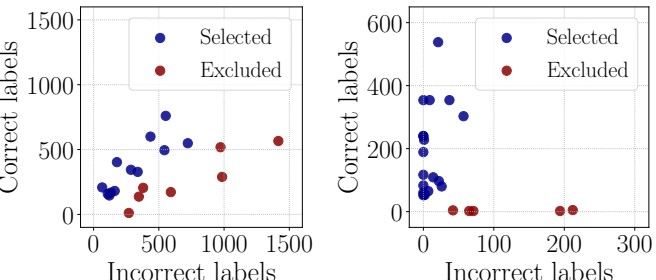

Figure 5: We find that the selected and not-selected source tasks are separated in terms of the number of correct labels provided by each source task versus the number of incorrect labels from each source task. Each dot represents one labeling function.

Table 2: Worst-group accuracies using MTL with source tasks selected by our algorithm, as compared to STL, MTL optimization methods, and exhaustive search over combinations of up to two source tasks.

| Dataset | HI | KS | LA | NJ | NV | SC |
|---|---|---|---|---|---|---|
| Training | 4,638 | 9,484 | 12,400 | 28,668 | 8,884 | 14,927 |
| Validation | 1,546 | 3,161 | 4,133 | 9,556 | 2,961 | 4,976 |
| Test | 1,547 | 3,162 | 4,134 | 9,557 | 2,962 | 4,976 |
| Smallest group size | 67 | 75 | 58 | 52 | 61 | 203 |
| GroupDRO | 74.56±0.58 | 75.50±0.59 | 74.90±0.38 | 76.95±0.20 | 73.06±0.66 | 75.56±1.36 |
| Correct-n-contrast | 74.37±0.27 | 75.52±1.19 | 74.25±0.15 | 77.60±0.10 | 73.22±0.40 | 76.23±0.98 |
| Naive MTL | 73.63±0.46 | 75.22±0.73 | 73.24±1.01 | 77.28±0.25 | 73.22±1.12 | 76.23±0.49 |
| HOA | 74.67±0.32 | 75.22±1.48 | 73.69±0.86 | 77.49±0.25 | 73.88±0.66 | 76.80±0.65 |
| TAG | 74.48±0.41 | 75.97±1.18 | 73.24±1.01 | 77.41±0.48 | 74.05±0.84 | 76.41±0.50 |
| TAWT | 73.53±0.44 | 75.14±1.39 | 73.51±1.38 | 76.47±1.31 | 72.89±0.81 | 76.59±0.97 |
| Exhaustive search ($\alpha \leq 2$) | 75.10±0.37 | **77.03±0.76** | 73.60±1.02 | 77.40±0.24 | 73.21±1.10 | 77.16±0.21 |
| **Alg. 1 (Ours)** | **75.47±0.73** | 76.96±0.69 | **75.62±0.11** | **78.17±0.36** | **75.21±0.52** | **77.62±0.34** |

**NLP tasks.** Next, we test our approach to NLP tasks. We collect 25 datasets from GLUE, SuperGLUE, TweetEval, and ANLI. See Table 4, Appendix C.2 for a complete list. We evaluate our approach by first selecting source tasks and then applying MTL. We test on five target tasks: CoLA, RTE, CB, COPA, and WSC. For each task, we use the rest 24 tasks as source tasks.

We first compare our approach with STL and naive MTL. We observe that naive MTL can perform worse than STL, e.g., on CoLA and WSC datasets. By contrast, our approach always outperforms STL (by **5.5%**) and naive MTL (by **5.4%**), on average. We then compare our approach with TAG and HOA. Our approach shows an average improvement of **2.2%** and is especially effective for tasks with a small training set.

### 4.4 Results for Improving Robustness in Multi-group Learning

We apply our approach to multi-group learning settings where the input distribution contains a heterogeneous mixture of subpopulations. The objective of these problems is to learn a model that performs robustly for all groups. In particular, we apply our approach to three performance metrics: worst-group accuracy, democratic disparity, and equality of opportunity. We also compare against STL methods, including group distributional robust optimization (GroupDRO, Sagawa et al. (2020)) and supervised contrastive learning (correct-n-contrast, Zhang et al. (2022)). Table 2 presents the comparison.

Compared with single-task learning, including GroupDRO and correct-n-contrast, we find that task modeling improves the worst-group accuracy by **1.17%** on average. Compared with existing MTL optimization methods, our approach shows a favorable gain of up to **1.9%** absolute accuracy. Measured by two fairness metrics, namely democratic disparity and equality of opportunity, our algorithm outperforms all baseline methods by **1.8%** on average. These results can be found in Table 6 of Appendix C.2.

### 4.5 Techniques to Speed Up Training

Lastly, we show that we can further reduce the runtime of our approach by adding two techniques. Our objective is to achieve comparable performance results to the ones shown in Table 1, but we will speed up the computation of $f(S_1), f(S_2), \ldots, f(S_n)$ using the following two simple techniques.

- First, we can reduce the size of the training set for computing $f$ by downsampling the training data from each task by a fixed proportion.

- Second, we can reduce the number of iterations for training each MTL model by early stopping the training procedure.

To illustrate the benefit of these two techniques, we apply them to two weak supervision datasets. The results are shown in Table 3. We find that by downsampling **40%** of the training data and early stopping at **20%** of the training epochs, we can achieve comparable performance to fully training MTL models. In particular, the

Table 3: Speeding up our approach by training models on sampled subsets of tasks with 20% training epochs (early stopping) and 40% training data (downsampling). With these two speed-up techniques, we can achieve comparable performance compared to fully computing the MTL performances.

| Dataset (Metrics) | CDR (Hours / F1) | Chemprot (Hours / Acc.) |
|---|---|---|
| Alg. 1 w/o early stopping and downsampling | 38.34H / 61.22±0.39 | 31.14H / 57.54±0.55 |
| Alg. 1 w/ early stopping and downsampling | 2.89H / 60.77±0.05 | 3.76H / 57.06±0.84 |

accuracy difference is within 0.5% for both datasets. However, we manage to reduce the training time for computing $f(S_1), f(S_2), \ldots, f(S_n)$ by **12×** times. In Table 7 of Appendix C.3, we report the running time for all the baselines on these two datasets. Overall, our approach is comparable to the baseline optimization methods in terms of efficiency.

### 4.6 Ablation Studies

**Benefit of modeling higher-order transfers.** We validate the benefit of modeling higher-order task transfers over approaches that only precompute first-order or second-order task affinities. First, compared with approaches that compute first-order task affinities, our approach improves the accuracy by **3.0%**, as is clear from Tables 1 and 2. Second, we precompute the MTL performance for every combination of two source tasks. We run an exhaustive search over $k(k-1)/2$ combinations to find the best combination for MTL. We test on six target tasks with $k = 50$, which requires training $1,225$ MTL models with two source tasks and one target task each time. We find that our selection procedure consistently outperforms the best two-task subsets by **1.21%** absolute accuracy. This is shown on the last two lines in Table 2.

**Sensitivity of model parameters.** There are three parameters that require tuning: the subset size $\alpha$, the number of samples $n$, and the loss function $\ell$. We vary $\alpha$ for each dataset between $\{3, 5, 10, 15\}$ via cross-validation, on a holdout set of 100 subsets. We pick $n$ in $\{50, 200, 400, 800\}$ according to the number of tasks $k$. Besides, we find that choosing $\ell$ as the classification margin function performs the best in practice.

The threshold $\gamma$ is usually set as 0.3 or 0.4 for weak supervision datasets, which selects most of the source tasks on average except the highly noisy labels. For instance, on the Semeval dataset with 164 source tasks, our approach selected 160, while $\alpha$ is 15. For the NLP and multi-group learning tasks, $\gamma$ is usually set as $-0.5$. This usually selects 3 or 4 source tasks, while $\alpha$ is 5. Thus, there is only a small number of helpful source tasks for a particular target task.

Lastly, we find that the selected tasks remain the same when using different random seeds to train the surrogate model. For details, see Appendix C.4.

## 5 Related Work

We note that there is a vast body of work on multi-task learning from various fields. A recurring theme for multitask learning research is inspired by a desire to imitate human intelligence as we continue to learn new information and extrapolate the learned information to new tasks and domains (Thrun and Pratt, 1998). In the early literature, many studies focus on MTL with linear and kernel-based models. A common approach is to set up separate parameters for each task while adding explicit regularization to the combined parameters (Evgeniou and Pontil, 2004; Argyriou et al., 2007; 2008). For linear models, this approach can be related to low-rank matrix approximation (Ando and Zhang, 2005). Inspired by the development of deep learning, recent works focus on MTL with deep neural networks (Yang and Hospedales, 2017). More broadly, see several recent surveys (Zhang and Yang, 2021; Jiang et al., 2022) for more comprehensive references. Within this vast literature, the contribution of our work is in the identification of negative transfers and the design of subset selection methods. Below, we discuss several relevant topics in detail.

**Understanding Black-box Predictions.** Surrogate modeling is a classic technique for studying black-box functions (Sacks et al., 1989; Ong et al., 2003), which we use as a proxy to study task relatedness. Our approach builds on the recent work of datamodels (Ilyas et al., 2022). However, there are two major differences between our work and their work. First, we apply the idea of surrogate models to multitask learning, whereas

their work focuses on the single-task supervised learning setting. Second, besides empirical demonstrations, we have also conducted a theoretical analysis of our approach to multi-task learning. Our findings reinforce the result of Ilyas et al. (2022) that the performances of deep neural networks can be extrapolated efficiently and accurately. Recent work has sought to explain why datamodels can perform well using harmonic analysis (Saunshi et al., 2022). It would be interesting to see if their techniques can be used to explain the empirical findings of our work in the context of MTL. More broadly, there is a line of work on developing techniques to understand the influence of data in black-box models through influence functions. See Koh and Liang (2017) and Yeh et al. (2018) for further references.

**Formal Notions of Task-relatedness.** There is a rich discussion about formulating notions of task-relatedness in the literature (Ben-David and Schuller, 2003). Ben-David et al. (2010) introduces a discrepancy notion called $\mathcal{H}$-divergence, which leads to a generalization bound for minimizing the empirical risk of naive MTL. Transfer exponents are another measure of discrepancy between two distributions (Hanneke and Kpotufe, 2019). Geometric distance measures for linear data models have also been considered in few-shot learning (Du et al., 2020) and meta-learning (Kong et al., 2020; Saunshi et al., 2021).

Note that none of these task-relatedness measures can be measured on deep neural networks due to the complexity of these models. One heuristic solution is to measure the cosine similarity between the gradients of each task's loss functions during training (Yu et al., 2020; Dery et al., 2021; Chen et al., 2022). Another solution is to measure the similarity of the predicted probabilities between tasks (Nguyen et al., 2020). This leads to a noisy estimate of task-relatedness, which is best for capturing first-order transfers. Standley et al. (2020) combines domain knowledge from visual intelligence to build a task relation taxonomy for 26 tasks. Compared with their approach, our approach is more generic, applies to any MTL settings with little to no domain knowledge, and efficiently captures higher-order transfer in a principled framework. Rather than defining an explicit relatedness measure, our work uses surrogate models to measure task-relatedness. This perspective circumvents the design of explicit task-relatedness measures for deep neural networks but is still useful for predicting transfers and for optimizing the performance of MTL.

**Optimization Methods for MTL.** An empirical motivation for this paper stems from recent work using weak supervision for training deep models (Ratner et al., 2016). We build on a multi-task weak supervision approach (Ratner et al., 2019) while adding new capability to deal with conflicts between labeling functions in the end model. This problem has also been studied in the rich literature about learning from noisy labels (Liu and Tao, 2015). For example, Xia et al. (2019) and Xia et al. (2020) propose to estimate transition matrices for multi-class prediction and use statistically-consistent weighting to integrate multiple noisy labels. Complementary to these works, we fit a surrogate model to approximate multitask learning performances and use the surrogate model to predict the performance of unseen task combinations.

Our approach selects source tasks for learning a target task, which has been studied in several recent works using optimization methods (Guo et al., 2019; Chen et al., 2022). Recent work (Liu et al., 2022) optimizes a weighted combination of per-task loss functions and jointly updates task-specific weights by the gradients of per-task losses during training. By contrast, our approach focuses on subset selection. Besides, our approach can separate tasks with more noisy labels when source tasks have disparate labeling precision. Our setting is also related to the task grouping problem (Kumar and Daume III, 2012), which aims to assign tasks into several groups, with each group of tasks learned in a separate MTL model. Unlike this problem, we select a subset of source tasks for a particular target task.

There are also works that apply low-rank tensor factorization to the parameters of multiple linear regression tasks (Wimalawarne et al., 2014). Along this line of research, several recent works apply low-rank regularization methods with a block-diagonal structure on the model parameters (Nie et al., 2018; Yang et al., 2020). Yang and Hospedales (2017) revisit the idea of tensor factorization in the context of deep neural networks. Liu et al. (2016) provide generalization bounds for multi-task learning under a low-rank structural condition on all the tasks. Their results shed light on when MTL would be better than STL.

Lastly, we note that task relations are characteristically different between different benchmarks due to the nature of the data. This paper focuses on developing a methodology for predicting MTL performances using rigorous theoretical and empirical arguments. Our extensive experiments demonstrate the usefulness of the methodology. It would be interesting to apply our methodology to large-scale benchmarks beyond what

we have studied (Zamir et al., 2018; Aribandi et al., 2022). Besides, it would be interesting to see if our approach can be applied to other related settings such as federated learning (Wang et al., 2020) and multitask reinforcement learning (Wang et al., 2022), where the problem of identifying negative transfers also arises. Lastly, although our work focuses on subset selection for multitask learning at the task level, it would be interesting to see if similar approaches could be applied at the feature level.

## 6 Conclusion

This paper studied how to efficiently predict negative transfers from multiple source tasks to one target task. The main contribution is the design and analysis of surrogate models for predicting multi-task learning performances. Both theoretical and empirical results show that our approach is efficient, accurate, and advances over prior optimization methods for multi-task learning.

Our work opens up many interesting questions for future work. Although we demonstrated the empirical strength of linear models for MTL, a rigorous explanation is lacking; Can recent analytic tools for understanding datamodels (Saunshi et al., 2022) be used here? Can more advanced sampling techniques, such as adaptive sampling, help speed up the training of surrogate models, which might enable the training of more powerful models? Lastly, our experiments show that the validation set size of the target task does not need to be very large for the approach to perform well. This is currently not explained by our Rademacher complexity-based bound. It is possible that with a tighter generalization analysis via data-dependent bounds, one might get a result that captures few-shot learning scenarios. This would be an interesting question for future work. Understanding task relationships in multitask learning is a complex and challenging research question. We hope our work inspires more principled studies in this direction.

### Acknowledgment

Thanks to Andrew Ilyas, Simon Du, Shuxiao Chen, and Chicheng Zhang for helpful discussions at various stages of this work. Thanks to the anonymous referees and the action editor for providing constructive feedback on our work. DL acknowledges the financial support from a seed grant and the startup fund from the Khoury College of Computer Sciences, Northeastern University.

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

# A    Results for Constructing Surrogate Models

Section 2.3 shows that the sample complexity for learning task models is linear in the number of source tasks. Here, we provide empirical evidence to support this result. We plot the convergence of task modeling on sixteen target tasks from three datasets described in Section 4.1. We measure the MSE between task model predictions and empirical training results on the holdout set of size 100, following the experimental setup described in Section 4.2. Figure 6 shows the results. The red line shows the variance of $f$, measured across five random seeds. We observe that:

- The MSE of task models consistently converges close to the variance of the prediction loss.
- The Spearman correlation coefficient between the predictions and the true performances is **0.8** on average.

Thus, we conclude that linear surrogate models can be accurately fitted with less than $8k$ samples, and the fitted model can accurately predict the performances of unsampled subsets.

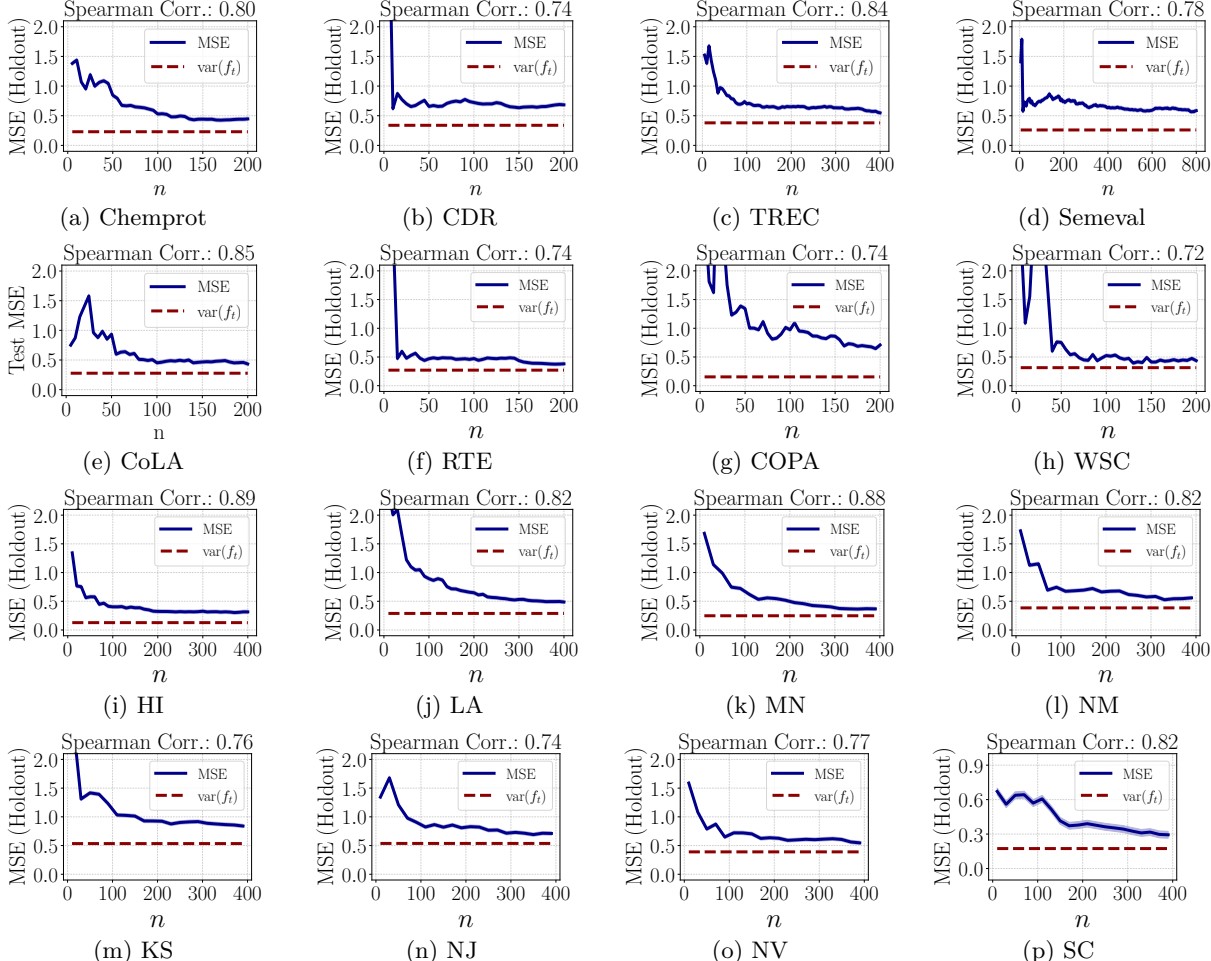

Figure 6: The MSE of linear surrogate models converges close to the variance of MTL performances. **(a-d)** Weak supervision tasks. **(e-h)** NLP tasks. **(i-p)** Multi-group learning tasks.

# B    Complete Proofs of Theoretical Results

**Notations.** We use $f(n) = (1 + o(1))g(n)$ to indicate that $|f(n) - g(n)|/g(n)$ approaches zero as $n$ goes to infinity. For a matrix denoted as $X$, denote the spectral norm (or the largest singular value) of $X$ as $\|X\|_2$. Denote the Frobenius norm of $X$ as $\|X\|_F$.

### B.1 Proof of Lemma 2.2

In the first part of the proof, we prove the convergence from $\hat{\theta}$ to $\bar{\theta}$ by dealing with the randomness of $S_1, S_2, \ldots, S_n$. Recall that $\mathcal{U}$ is the uniform distribution over subsets of $\{1, 2, \ldots, k\}$ with size $\alpha$. Let $|\mathcal{U}| = \binom{k}{\alpha}$ denote the number of subsets from $\mathcal{U}$.

*Proof of Lemma 2.2.* Recall the definitions of $\hat{\theta}$ and $\bar{\theta}$ from Section 2. They can be written equivalently as follows:

$$\hat{\theta} = \left( \frac{\mathcal{I}_n^\top \mathcal{I}_n}{n} \right)^{-1} \frac{v}{n} \quad \text{and} \quad \bar{\theta} = \left( \frac{\boldsymbol{\mathcal{I}}^\top \boldsymbol{\mathcal{I}}}{|\mathcal{U}|} \right)^{-1} \frac{\boldsymbol{\mathcal{I}}^\top \boldsymbol{f}}{|\mathcal{U}|}.$$

We will use the triangle inequality to separate the error between $\hat{\theta}$ and $\bar{\theta}$ into two parts:

$$\left\| \hat{\theta} - \bar{\theta} \right\| = \left\| \left( \left( \frac{\mathcal{I}_n^\top \mathcal{I}_n}{n} \right)^{-1} - \left( \frac{\boldsymbol{\mathcal{I}}^\top \boldsymbol{\mathcal{I}}}{|\mathcal{U}|} \right)^{-1} \right) \frac{v}{n} + \left( \frac{\boldsymbol{\mathcal{I}}^\top \boldsymbol{\mathcal{I}}}{|\mathcal{U}|} \right)^{-1} \left( \frac{v}{n} - \frac{\boldsymbol{\mathcal{I}}^\top \boldsymbol{f}}{|\mathcal{U}|} \right) \right\|$$

$$\leq \left\| \left( \frac{\mathcal{I}_n^\top \mathcal{I}_n}{n} \right)^{-1} - \left( \frac{\boldsymbol{\mathcal{I}}^\top \boldsymbol{\mathcal{I}}}{|\mathcal{U}|} \right)^{-1} \right\|_2 \cdot \left\| \frac{v}{n} \right\| \tag{14}$$

$$+ \left\| \left( \frac{\boldsymbol{\mathcal{I}}^\top \boldsymbol{\mathcal{I}}}{|\mathcal{U}|} \right)^{-1} \right\|_2 \cdot \left\| \frac{v}{n} - \frac{\boldsymbol{\mathcal{I}}^\top \boldsymbol{f}}{|\mathcal{U}|} \right\|. \tag{15}$$

We compare $\frac{v}{n}$ and $\frac{\boldsymbol{\mathcal{I}}^\top \boldsymbol{f}}{|\mathcal{U}|}$. Recall that both vectors have $k$ coordinates, each corresponding to one task. For any task $i = 1, \ldots, k$, let $\mathcal{E}_i$ denote the difference between the $i$-th coordinate of $\frac{v}{n}$ and the $i$-th coordinate of $\frac{\boldsymbol{\mathcal{I}}^\top \boldsymbol{f}}{|\mathcal{U}|}$:

$$\mathcal{E}_i = \frac{1}{n} \sum_{1 \leq j \leq n: \ i \in S_j} f(S_j) - \frac{1}{|\mathcal{U}|} \sum_{T \in \mathcal{U}: \ i \in T} f(T). \tag{16}$$

Notice that the sampling of $S_1, S_2, \ldots, S_n$ is independent of the randomness in $f$. Therefore, we have that the expectation of $\mathcal{E}_i$ is zero:

$$\mathbb{E}[\mathcal{E}_i] = 0, \text{ for any } i = 1, 2, \ldots, k.$$

Next, we apply Chebyshev's inequality to analyze the deviation of $\mathcal{E}_i$ from its expectation. We consider the variance of $\mathcal{E}_i$, which is equal to the expectation of $\mathcal{E}_i^2$ since the mean of $\mathcal{E}_i$ is zero:

$$\mathbb{E}\left[\mathcal{E}_i^2\right] = \mathbb{E}\left[ \left( \frac{1}{n} \sum_{1 \leq j \leq n: \ i \in S_j} f(S_j) - \frac{1}{|\mathcal{U}|} \sum_{T \in \mathcal{U}: \ i \in T} f(T) \right)^2 \right]$$

$$= \mathbb{E}\left[ \frac{1}{n^2} \left( \sum_{1 \leq j \leq n: i \in S_j} f(S_j) \right)^2 - \frac{2}{n |\mathcal{U}|} \sum_{1 \leq j \leq n: \ i \in S_j} f(S_j) \sum_{T \in \mathcal{U}: \ i \in T} f(T) + \frac{1}{|\mathcal{U}|^2} \left( \sum_{T \in \mathcal{U}: \ i \in T} f(T) \right)^2 \right] \tag{17}$$

Notice that for any $T \in \mathcal{U}$ such that $i \in T$, the probability that $T$ is sampled in the training dataset of size $n$ is equal to

$$\frac{\binom{|\mathcal{U}|-1}{n-1}}{\binom{|\mathcal{U}|}{n}} = \frac{n}{|\mathcal{U}|}.$$

For any two subsets $T \neq T'$ that are both from $\mathcal{U}$ such that $i \in T$ and $i \in T'$, the probability that $T$ and $T'$ are both sampled in the training set (of size $n$) is equal to

$$\frac{\binom{|\mathcal{U}|-1}{n-1}}{\binom{|\mathcal{U}|}{n}} \cdot \frac{\binom{|\mathcal{U}|-1}{n-1}}{\binom{|\mathcal{U}|}{n}} = \frac{n^2}{|\mathcal{U}|^2}.$$

Thus, by taking the expectation over the randomness of the sampled subsets in equation (17) conditional on $f$, we can cancel out the cross terms for every pair of two tasks $i \neq i'$, leaving only the squared terms as:

$$\mathbb{E}\left[\mathcal{E}_i^2\right] = \mathbb{E}\left[\left(\frac{1}{n^2}\frac{n}{|\mathcal{U}|} - \frac{2}{n}\frac{n}{|\mathcal{U}|}\frac{1}{|\mathcal{U}|} + \frac{1}{|\mathcal{U}|^2}\right)\sum_{T \in \mathcal{U}:\, i \in T}\left(f(T)\right)^2\right] \leq \frac{C^2}{n} \cdot \frac{|T \in \mathcal{U}:\, i \in T|}{|\mathcal{U}|} \leq \frac{C^2}{n},$$

since the value of $f$ is bounded from above by an absolute constant $C$. Therefore,

$$\mathbb{E}\left[\sum_{i=1}^{k}\mathcal{E}_i^2\right] \leq \frac{C^2 k}{n}.$$

By Markov's inequality, for any $a > 0$,

$$\Pr\left[\sum_{i=1}^{k}\mathcal{E}_i^2 \geq \frac{a^2 k}{n}\right] \leq \frac{C^2}{a^2}.$$

Therefore, with probability at least $1 - \delta$, for any $\delta > 0$, conditional on the randomness of $f$, we have that

$$\left\|\frac{v}{n} - \frac{\boldsymbol{\mathcal{I}}^\top \boldsymbol{f}}{|\mathcal{U}|}\right\| \leq C\sqrt{\frac{k}{\delta n}}. \tag{18}$$

Next, we use random matrix concentration results to analyze the difference between the indicator matrix of the sampled subsets and the indicator matrix of all subsets in $\mathcal{U}$. Denote by

$$E = \frac{\mathcal{I}_n^\top \mathcal{I}_n}{n} - \frac{\boldsymbol{\mathcal{I}}^\top \boldsymbol{\mathcal{I}}}{|\mathcal{U}|} \quad \text{and} \quad A = \frac{\boldsymbol{\mathcal{I}}^\top \boldsymbol{\mathcal{I}}}{|\mathcal{U}|}.$$

By the Sherman-Morrison formula calculating matrix inversions, we get

$$\begin{aligned}
\left\|\left(\frac{\mathcal{I}_n^\top \mathcal{I}_n}{n}\right)^{-1} - \left(\frac{\boldsymbol{\mathcal{I}}^\top \boldsymbol{\mathcal{I}}}{|\mathcal{U}|}\right)^{-1}\right\|_2 &= \left\|(E + A)^{-1} - A^{-1}\right\|_2 \\
&= \left\|A^{-1}\left(AE^{-1} + \mathrm{Id}_{k \times k}\right)^{-1}\right\|_2 \\
&= \left\|A^{-1}E\left(A + E\right)^{-1}\right\|_2 \\
&\leq \left(\lambda_{\min}(A)\right)^{-1} \cdot \|E\|_2 \cdot \left(\lambda_{\min}(A + E)\right)^{-1} \\
&\leq \frac{\|E\|_2}{\lambda_{\min}(A)(\lambda_{\min}(A) - \|E\|_2)}.
\end{aligned} \tag{19}$$

We now use the matrix Bernstein inequality (cf. Theorem 6.1.1 in Tropp (2015)) to deal with the spectral norm of $E$. Let

$$X_i = \mathbb{1}_{S_i}\mathbb{1}_{S_i}^\top - \frac{\boldsymbol{\mathcal{I}}^\top \boldsymbol{\mathcal{I}}}{|\mathcal{U}|}, \quad \text{for any} \ \ i = 1, \dots, n.$$

In expectation over $\mathcal{U}$, we know that $\mathbb{E}\left[X_i\right] = 0$, for any $i = 1, \dots, n$. Additionally, $\|X_i\|_2 \leq 2\alpha$, since it is a linear combination of indicator vectors with $\alpha$ entries of ones in each indicator vector. Therefore, for all $t \geq 0$,

$$\Pr\left[\|E\|_2 \geq t\right] = \Pr\left[\left\|\sum_{i=1}^{n}X_i\right\|_2 \geq nt\right] \leq 2k \cdot \exp\left(-\frac{(nt)^2/2}{(2\alpha)^2 n + (2\alpha)nt/3}\right).$$

With some standard calculations, this implies that for any $\delta \geq 0$, with probability at least $1 - \delta$,

$$\|E\|_2 \leq \frac{4\alpha \cdot \log\left(2k\delta^{-1}\right)}{\sqrt{n}}. \tag{20}$$

By applying equation (18) into equation (14) and equation (20) into equation (15), we have shown that with probability at least $1 - 2\delta$, for any $\delta \geq 0$,

$$\left\| \hat{\theta} - \bar{\theta} \right\| \leq \left\| \frac{v}{n} \right\|_2 \cdot \frac{4\alpha \cdot \log\left(2k\delta^{-1}\right)}{\sqrt{n}} + \frac{1}{\left(\lambda_{\min}(A)\right)^2 \left(\lambda_{\min}(A) - \|E\|_2\right)} \cdot C\sqrt{\frac{k}{\delta n}}. \tag{21}$$

Lastly, we examine the norm of $\frac{v}{n}$. Let $z_i$ be the number of subsets $S_j$ among $1 \leq j \leq n$ such that $i \in S_j$, for any $i = 1, \dots, n$. Recall that the value of $f$ is bounded from above by an absolute constant $C$. Thus, based on the definition of $v$ from equation (8), we have:

$$\left\| \frac{v}{n} \right\| \leq \frac{1}{n} \sqrt{C^2 \sum_{i=1}^{k} z_i^2} \leq \frac{C}{n} \left( \sum_{i=1}^{k} z_i \right) = C\alpha, \tag{22}$$

since the size of each subset is strictly equal to $\alpha$.

Regarding the minimum eigenvalue of $A$, notice that the diagonal entry of $\frac{\mathcal{I}^\top \mathcal{I}}{|\mathcal{U}|}$ is equal to $\binom{k-1}{\alpha-1}$. The off-diagonal entries of this matrix are equal to $\binom{k-2}{\alpha-2}$. Thus, based on standard algebra, one can prove that

$$\lambda_{\min}(A) \geq 1 - \frac{\binom{k-2}{\alpha-2}}{\binom{k-1}{\alpha-1}} = 1 - \frac{\alpha-1}{k-1} \geq 1 - \frac{\alpha}{k}. \tag{23}$$

Applying equations (22) and (23) back into equation (21), we conclude that with probability at least $1 - 2\delta$, $\hat{\theta}$ the estimation error between $\hat{\theta}$ and $\bar{\theta}$ grows at a rate of $\sqrt{\frac{k}{n}}$ as follows:

$$\left\| \hat{\theta} - \bar{\theta} \right\| \leq 4C\alpha^2 \log(2k\delta^{-1}) \cdot \sqrt{\frac{k}{n}} + \left(1 - \frac{\alpha}{k}\right)^{-3} C\alpha \cdot \sqrt{\frac{k}{\delta n}}.$$

Thus, we have proved that equation (9) holds, and the proof is complete. $\qquad \square$

### B.2 Proof of Lemma 2.3

In the second part, we prove the convergence from $\bar{\theta}$ to $\theta^\star$ by dealing with the randomness of $f$.

*Proof of Lemma 2.3.* Based on the definitions of $\bar{\theta}$ and $\theta^\star$, their difference can be written as follows:

$$\left\| \bar{\theta} - \theta^\star \right\| = \left\| \left( \mathcal{I}^\top \mathcal{I} \right)^{-1} \mathcal{I}^\top \left( \boldsymbol{f} - \mathbb{E}\left[\boldsymbol{f}\right] \right) \right\| \tag{24}$$

$$\leq \left\| \left( \frac{\mathcal{I}^\top \mathcal{I}}{|\mathcal{U}|} \right)^{-1} \frac{\mathcal{I}^\top}{\sqrt{|\mathcal{U}|}} \right\|_2 \cdot \left\| \frac{\boldsymbol{f} - \mathbb{E}\left[\boldsymbol{f}\right]}{\sqrt{|\mathcal{U}|}} \right\|$$

$$= \sqrt{\left( \frac{\mathcal{I}^\top \mathcal{I}}{|\mathcal{U}|} \right)^{-1}} \cdot \frac{\|\boldsymbol{f} - \mathbb{E}\left[\boldsymbol{f}\right]\|}{\sqrt{|\mathcal{U}|}}$$

$$\leq \left( 1 - \frac{\alpha}{k} \right)^{-\frac{1}{2}} \cdot \frac{\|\boldsymbol{f} - \mathbb{E}\left[\boldsymbol{f}\right]\|}{\sqrt{|\mathcal{U}|}}. \qquad \text{(by equation (23))}$$

For each subset $T \in \mathcal{U}$, recall that $f(T)$ is the MTL outcome of combining the datasets of all tasks of $T$ with the main target task. We will apply a Rademacher complexity-based generalization bound to analyze the generalization error $f(T) - \mathbb{E}\left[f(T)\right]$. Recall the Rademacher complexity of $\mathcal{F}$ with $m$ samples from $\mathcal{D}_t$ is defined in equation (5). By Bartlett and Mendelson (2002, Theorem 5), with probability at least $1 - \delta$, we can get:

$$f(T) \leq \mathbb{E}\left[f(T)\right] + \frac{\mathcal{R}_m(\mathcal{F})}{2} + \sqrt{\frac{\log\left(1/\delta\right)}{2m}}. \tag{25}$$

Similarly, one can get the result for the other directions of the error estimate. With a union bound over all subsets $T \in \mathcal{U}$, with probability at least $1 - \delta$, we get:

$$f(T) \leq \mathbb{E}\left[f(T)\right] + \frac{\mathcal{R}_m(\mathcal{F})}{2} + \sqrt{\frac{\alpha \log\left(\frac{k}{\delta}\right)}{2m}}, \text{ for all } T \in \mathcal{U}, \tag{26}$$

since

$$\log\left(\frac{\binom{k}{\alpha}}{\delta}\right) \leq \alpha \log\left(\frac{k}{\delta}\right).$$

Let $z = \sqrt{\alpha \log\left(k\delta^{-1}\right)/(2m)}$. Applying equation (26) back into equation (24), we have shown

$$\left\|\bar{\theta} - \theta^\star\right\| \leq \left(1 - \frac{\alpha}{k}\right)^{-\frac{1}{2}} \sqrt{\frac{1}{|\mathcal{U}|}\sum_{T \in \mathcal{U}}\left(\frac{\mathcal{R}_m(\mathcal{F})}{2} + z\right)^2}$$

$$= \left(1 - \frac{\alpha}{k}\right)^{-\frac{1}{2}}\left(\frac{\mathcal{R}_m(\mathcal{F})}{2} + z\right).$$

Thus, based on the condition that $\alpha \leq k/2$, the proof of equation (10) is complete. $\square$

*Proof of Theorem 2.1.* Notice that equation (6) follows by combining equation (9) from Lemma 2.2 and equation (10) from Lemma 2.3, together with the condition that $\alpha \leq 1/2$. Thus, the proof of the theorem is finished. $\square$

**Remark.** Our result depends on the Rademacher complexity of the function class. This complexity measure can be vacuous on real data for deep neural networks. It would be interesting to incorporate data-dependent generalization bounds in the proof (e.g., Li and Zhang (2021) and Ju et al. (2022), 2023).

### B.3 Convergence of the Empirical Risk

Based on the results from Lemma 2.2 and Lemma 2.3, we can also prove the convergence of the loss values. This is stated precisely in the following result.

**Corollary B.1** (of Theorem 2.1)**.** *In the setting of Theorem 2.1, we have that*

$$\mathcal{L}(\theta^\star) - \hat{\mathcal{L}}_n(\hat{\theta}) \lesssim C\alpha \cdot \mathcal{R}_m(\mathcal{F}) + C\alpha^{1.5}\sqrt{\frac{\log(\delta^{-1}k)}{m}} + C^2\alpha^{3.5}\log\left(\frac{k}{\delta}\right)\sqrt{\frac{k}{n}} + C^2\alpha^{2.5}\sqrt{\frac{k}{\delta n}}. \tag{27}$$

*Proof.* To analyze the generalization error of $\hat{\theta}$, based on equation (2), we can expand the loss term as

$$\hat{\mathcal{L}}_n(\hat{\theta}) = \frac{1}{n}\left\|\mathcal{I}_n\hat{\theta} - \hat{f}\right\|^2$$

$$= \frac{1}{n}\left\|\mathcal{I}_n\hat{\theta} - \mathbb{E}_{\hat{f}}\left[\hat{f}\right] + \mathbb{E}_{\hat{f}}\left[\hat{f}\right] - \hat{f}\right\|^2$$

$$= \frac{1}{n}\left\|\mathcal{I}_n\hat{\theta} - \mathbb{E}_{\hat{f}}\left[\hat{f}\right]\right\|^2 + \frac{2}{n}\langle\mathcal{I}_n\hat{\theta}_n - \mathbb{E}_{\hat{f}}\left[\hat{f}\right], \mathbb{E}_{\hat{f}}\left[\hat{f}\right] - \hat{f}\rangle + \frac{1}{n}\left\|\mathbb{E}_{\hat{f}}\left[\hat{f}\right] - \hat{f}\right\|^2. \tag{28}$$

Based on Lemma 2.2, the distance between $\hat{\theta}$ and $\theta^\star$ is at the order of $\mathrm{O}(n^{-1/2})$ with high probability. We will use this result to deal with the first term in equation (28) as follows:

$$\frac{1}{n}\left\|\mathcal{I}_n\hat{\theta}_n - \mathop{\mathbb{E}}_{\hat{f}}\left[\hat{f}\right]\right\|^2 - \frac{1}{n}\left\|\mathcal{I}_n\theta^\star - \mathop{\mathbb{E}}_{\hat{f}}\left[\hat{f}\right]\right\|^2 \tag{29}$$

$$=\left|\frac{1}{n}\langle\mathcal{I}_n^\top\mathcal{I}_n, \hat{\theta}(\hat{\theta})^\top - \theta^\star(\theta^\star)^\top\rangle - \frac{2}{n}\langle\mathop{\mathbb{E}}_{\hat{f}}\left[\hat{f}\right], \hat{\theta} - \theta^\star\rangle\right|$$

$$\leq\left\|\frac{1}{n}\mathcal{I}_n^\top\mathcal{I}_n\right\|_2 \cdot \left\|\theta^\star(\theta^\star)^\top - \hat{\theta}(\hat{\theta})^\top\right\|_F + \frac{2}{n}\left\|\mathop{\mathbb{E}}_{\hat{f}}\left[\hat{f}\right]\right\| \cdot \left\|\theta^\star - \hat{\theta}\right\| \qquad \text{(by triangle inequality)}$$

$$\leq\alpha\left\|\theta^\star(\theta^\star)^\top - \hat{\theta}(\hat{\theta})^\top\right\|_F + 2C\alpha \cdot e_1,$$

where $e_1$ denotes the right hand side of equation (6). In the last step, the first part uses the fact that $\mathcal{I}_n^\top\mathcal{I}_n/n$ is the average of $n$ rank one matrix, each with spectral norm $\alpha$ since they have exactly $\alpha$ ones. The second part uses an argument similar to equation (22) and the result of equation (6). Next,

$$\left\|\theta^\star(\theta^\star)^\top - \hat{\theta}_n(\hat{\theta}_n)^\top\right\|_F = \left\|\theta^\star(\theta^\star - \hat{\theta}_n)^\top + (\theta^\star - \hat{\theta}_n)(\hat{\theta}_n)^\top\right\|_F$$

$$\leq\left\|\theta^\star(\theta^\star - \hat{\theta}_n)^\top\right\|_F + \left\|(\theta^\star - \hat{\theta}_n)(\hat{\theta})^\top\right\|_F \qquad \text{(by triangle inequality)}$$

$$\leq\left(\|\theta^\star\| + \left\|\hat{\theta}\right\|\right)e_1. \qquad \text{(by equation (6))}$$

We show that the norm of $\theta^\star$ and $\hat{\theta}_n$ are both bounded by a constant factor times $\sqrt{k}$. To see this,

$$\|\theta^\star\| = \left\|(\boldsymbol{\mathcal{I}}^\top\boldsymbol{\mathcal{I}})^{-1}\boldsymbol{\mathcal{I}}^\top\mathbb{E}[\boldsymbol{f}]\right\|$$

$$\leq\left\|\left(\frac{\boldsymbol{\mathcal{I}}^\top\boldsymbol{\mathcal{I}}}{|\mathcal{U}|}\right)^{-1}\right\|_2 \cdot \left\|\frac{\boldsymbol{\mathcal{I}}^\top\mathbb{E}[\boldsymbol{f}]}{|\mathcal{U}|}\right\|$$

$$\leq\left(1 - \frac{\alpha}{k}\right)^{-1} \cdot C\sqrt{\alpha} \qquad \text{(by equation (23) and the condition that } f \text{ is bounded by } C)$$

Notice that the spectral norm of the difference between $\boldsymbol{\mathcal{I}}^\top\boldsymbol{\mathcal{I}}/|\mathcal{U}|$ and $\mathcal{I}_n^\top\mathcal{I}_n/n$ has been analyzed in equation (20). Thus, with similar steps as above, we can show that

$$\left\|\hat{\theta}\right\| \leq \left(\left(1 - \frac{\alpha}{k}\right)^{-1} + \frac{4\alpha\log\left(2k\delta^{-1}\right)}{\sqrt{n}}\right)C\sqrt{k}.$$

To wrap up our analysis above, we have shown that equation (29) is at most

$$e_3 = \alpha\left(2(1 - \alpha/k)^{-1} + \frac{4\alpha\log\left(2k\delta^{-1}\right)}{\sqrt{n}}\right)C\sqrt{\alpha} \cdot e_1 + 2C\alpha \cdot e_1.$$

Next, we consider the second term in equation (28). Let $e_2 = \frac{\mathcal{R}_m(\mathcal{F})}{2} + \sqrt{\frac{\alpha\log(k/\delta)}{2m}}$ be the deviation error indicated in equation (26). Thus, every entry of $\hat{f} - \mathbb{E}\left[\hat{f}\right]$ is at most $e_2$. Besides, each entry of $\mathcal{I}_n\hat{\theta}_n - \mathbb{E}\left[\hat{f}\right]$ is less than

$$\sqrt{\alpha}\|\hat{\theta}_n\| + C,$$

because $\|\mathcal{I}_n\|_2 \leq \sqrt{\alpha}$ and $f$ is bounded from above by $C$. Thus, the second term in equation (28) is less than

$$e_4 = e_2\left(\sqrt{\alpha} \cdot \left(\left(1 - \frac{\alpha}{k}\right)^{-1} + \frac{4\alpha\log\left(2k\delta^{-1}\right)}{\sqrt{n}}\right)C\sqrt{\alpha} + C\right).$$

For the population loss $\mathcal{L}(\theta^\star)$, notice that

$$
\begin{aligned}
\mathcal{L}(\theta^\star) &= \mathop{\mathbb{E}}_{\boldsymbol{f}} \left[ \frac{1}{|\mathcal{U}|} \left\| \boldsymbol{\mathcal{I}}\theta^\star - \boldsymbol{f} \right\|^2 \right] \\
&= \mathop{\mathbb{E}}_{\boldsymbol{f}} \left[ \frac{1}{|\mathcal{U}|} \left\| \boldsymbol{\mathcal{I}}\theta^\star - \mathop{\mathbb{E}}_{\boldsymbol{f}} [\boldsymbol{f}] + \mathop{\mathbb{E}}_{\boldsymbol{f}} [\boldsymbol{f}] - \boldsymbol{f} \right\|^2 \right] \\
&= \frac{1}{|\mathcal{U}|} \left\| \boldsymbol{\mathcal{I}}\theta^\star - \mathop{\mathbb{E}}_{\boldsymbol{f}} [\boldsymbol{f}] \right\|^2 + \frac{1}{|\mathcal{U}|} \left( \mathop{\mathbb{E}}_{\boldsymbol{f}} \left[ \left\| \boldsymbol{f} - \mathop{\mathbb{E}}_{\boldsymbol{f}} [\boldsymbol{f}] \right\|^2 \right] \right)
\end{aligned}
\tag{30}
$$

We know that each entry of $\boldsymbol{\mathcal{I}}\theta^\star - \mathbb{E}[\boldsymbol{f}]$ is at most $(1 - \alpha/k)^{-1}\sqrt{\alpha} + C$. Thus, by Hoeffding's inequality, with probability at least $1 - \delta$, we have

$$
\left| \frac{1}{n} \left\| \mathcal{I}_n \theta^\star - \mathop{\mathbb{E}}_{\hat{f}} \left[ \hat{f} \right] \right\| - \frac{1}{|\mathcal{U}|} \left\| \boldsymbol{\mathcal{I}}\theta^\star - \mathop{\mathbb{E}}_{\boldsymbol{f}} [\boldsymbol{f}] \right\| \right| \le \left( (1 - \alpha/k)^{-1}\sqrt{\alpha} + C \right) \sqrt{\frac{\log\left(\delta^{-1}\right)}{n}}.
\tag{31}
$$

Lastly, we consider the third term in equation (28), compared with the second term in equation (30). For every $T \in \mathcal{U}$, let $e_T = f(T) - \mathbb{E}[f(T)]$. By equation (26), we know that $e_T$ is of order $O(m^{-1/2})$, for every $T \in \mathcal{U}$. Therefore

$$
\left| \frac{1}{n} \sum_{i=1}^{n} e_{S_i}^2 \right| \le \left( \frac{\mathcal{R}_m(\mathcal{F})}{2} + \sqrt{\frac{\alpha \log(k/\delta)}{2m}} \right)^2,
\tag{32}
$$

which is of order $O(m^{-1})$. Similarly, the same holds for the variance of $\boldsymbol{f}$ in the second term of equation (30).

Comparing equations (31) and (28), we have shown that

$$
\begin{aligned}
\mathcal{L}(\theta^\star) - \hat{\mathcal{L}}_n(\hat{\theta}) &\le \left( (1 - \alpha/k)^{-1}\sqrt{\alpha} + C + C^2 \right) \sqrt{\frac{\log(\delta^{-1})}{n}} + C \cdot e_2 + e_3 + e_4 \\
&\lesssim (C + C\alpha) \left( \mathcal{R}_m(\mathcal{F}) + \frac{\sqrt{\alpha \log(k\delta^{-1})}}{\sqrt{m}} \right) + \frac{C^2 \alpha^{7/2} \log\left(2k\delta^{-1}\right) + 8C^2 \alpha^{5/2}\delta^{-1/2}\sqrt{k}}{\sqrt{n}}.
\end{aligned}
$$

The above follows by incorporating the definitions of the error terms $e_2, e_3, e_4$. Thus, we have proved that equation (27) holds. The proof is now finished. $\square$

## B.4   Proof of Theorem 3.1

Recall that $\mathcal{I}_n \in \{0, 1\}^{n \times k}$ is the indicator matrix corresponding to the task indices from the training dataset. Given a set of tasks $S$ with size $\alpha$, denote their feature matrices and label vectors as $(X_1, Y_1), (X_2, Y_2), \ldots, (X_\alpha, Y_\alpha)$. With hard parameter sharing (Yang et al., 2021), we minimize

$$
\ell(B) = \sum_{i=1}^{\alpha} \| X_i B - Y_i \|^2.
\tag{33}
$$

The minimizer of $\ell(B)$, denoted as $\hat{B}$, is equal to the following

$$
\hat{B} = \left( \sum_{i=1}^{\alpha} X_i^\top X_i \right)^{-1} \left( \sum_{i=1}^{\alpha} X_i^\top Y_i \right).
$$

For isotropic covariates, by matrix concentration results, the loss of using $B$ on the validation set of the target task is equal to

$$
f(S) = \left\| \hat{B} - \beta^{(t)} \right\|^2 + O\left( \sqrt{\frac{p}{m}} \right).
$$

First, we state the proof of Lemma 3.3 from Section 3.1.

*Proof of Lemma 3.3.* We have that $Y_i = X_i \beta^{(i)} + \epsilon^{(i)}$, where $\epsilon^{(i)}$ is a random vector whose entries are sampled independently with mean 0 and variance $\sigma^2$. We have

$$f(S) = \left\| \left( \sum_{i=1}^{\alpha} X_i^\top X_i \right)^{-1} \sum_{i=1}^{\alpha} X_i^\top \epsilon^{(i)} \right\|^2. \tag{34}$$

For a task $i$, we know that its coefficient is equal to the $i$-th entry of

$$\left( \frac{\mathcal{I}_n^\top \mathcal{I}_n}{n} \right)^{-1} \frac{\mathcal{I}_n^\top \hat{f}}{n}.$$

Let $Z = \mathcal{I}_n^\top \mathcal{I}_n / n$. By equation (12), for any $i \neq j$, we observe that

$$
\begin{aligned}
\left| \frac{\hat{\theta}_i - \hat{\theta}_j}{n} - \frac{k}{\alpha} \cdot \frac{v_i - v_j}{n} \right| &= \left| (e_i - e_j)^\top \left( Z^{-1} - \mathbb{E}\left[ Z \right]^{-1} \right) \frac{v}{n} \right| \\
&\leq \| e_i - e_j \| \cdot \left\| Z^{-1} - \mathbb{E}\left[ Z \right]^{-1} \right\|_2 \cdot \left\| \frac{v}{n} \right\| \\
&\leq 2C\alpha \cdot \left\| Z^{-1} - \mathbb{E}\left[ Z \right]^{-1} \right\|_2 && \text{(by equation (22))} \\
&\leq \frac{4\alpha \log \left( 2k\delta^{-1} \right)}{\sqrt{n}} \frac{2}{(1 - \alpha/k)^2}. && \text{(by equations (19), (20), (23))}
\end{aligned}
$$

The last step follows by applying equations (20) and (23) into equation (19). Thus, we have finished the proof of equation (11). $\qquad \square$

Second, we show that provided $n$, and $d$ are sufficiently large, a separation exists in the coefficients of $v$ between good and bad tasks.

*Proof of Theorem 3.1.* We calculate $v_i / n$ for all $i = 1, \ldots, k$ and compare its value between a good task and a bad task. We first compare their expectations over the randomly sampled subsets. By equation (18), we get

$$\left| \frac{v_i}{n} - \frac{1}{|\mathcal{U}|} \sum_{T \in \mathcal{U}: \ i \in T} f(T) \right| \leq \frac{Ck\delta^{-1/2}}{\sqrt{n}}, \ \text{ and }$$

$$\left| \frac{v_j}{n} - \frac{1}{|\mathcal{U}|} \sum_{T \in \mathcal{U}: \ j \in T} f(T) \right| \leq \frac{Ck\delta^{-1/2}}{\sqrt{n}}.$$

Therefore, by applying the triangle inequality with the above two results, we get

$$\left| \frac{v_i - v_j}{n} - \frac{\sum_{T \in \mathcal{U}: i \in T} f(T) - \sum_{T \in \mathcal{U}: j \in T} f(T)}{|\mathcal{U}|} \right| \leq \frac{2Ck\delta^{-1/2}}{\sqrt{n}}. \tag{35}$$

To deal with equation (35), we apply a union bound over the sample covariance matrix of every subset $T$ in $\mathcal{U}$ to show that they are close to their expectation. By Gaussian covariance estimation results (e.g., Wainwright (2019, equation (6.12))), for a fixed $T \in \mathcal{U}$ such that $T = \{i_1, i_2, \ldots, i_\alpha\}$, we get

$$\left| \frac{1}{\alpha d} \sum_{j \in T} X_j^\top X_j - \text{Id}_{p \times p} \right| \leq 2\sqrt{\frac{p}{\alpha d}} + 2\epsilon + \left( \sqrt{\frac{p}{\alpha d}} + \epsilon \right)^2, \tag{36}$$

with probability at least $1 - 2\exp\left( -\frac{1}{2}\alpha d\epsilon^2 \right)$. With a union bound over all $T \in \mathcal{U}$, we have that the above holds with probability at least $1 - \delta$ for all $T \in \mathcal{U}$, for $\epsilon$ that is equal to

$$\epsilon = \sqrt{\frac{2\alpha k \log(2k\delta^{-1})}{\alpha d}}.$$

Let $\varepsilon_1$ denote the error term from equation (36), by inserting the value of $\epsilon$:

$$\varepsilon_1 = 2\sqrt{\frac{p}{\alpha d}} + 2\sqrt{\frac{2\alpha \log(2k\delta^{-1})}{\alpha d}} + \left(\sqrt{\frac{p}{\alpha d}} + \epsilon\right)^2.$$

Let

$$u_T = \frac{1}{\alpha d} \sum_{j \in T} X_j^\top \epsilon^{(j)}, \text{ for any } T \in \mathcal{U}.$$

One can verify that

$$\left| f(T) - \|u_T\|^2 \right| \le \left((1 - \varepsilon_1)^{-2} - 1\right) \|u_T\|^2 \le 3\varepsilon_1 \|u_T\|^2.$$

Notice that

$$\mathbb{E}\left[\|u_T\|^2\right] = \mathbb{E}\left[\frac{1}{(\alpha d)^2} \operatorname{Tr}\left[\sum_{j \in T} X_j^\top \varepsilon^{(j)} (\varepsilon^{(j)})^\top X_j\right]\right].$$

If $j$ is a good task, then the expectation over $\varepsilon^{(j)}$ is equal to $a^2 \operatorname{Id}$ by the assumption of Theorem 3.1. If $j$ is a bad task, on the other hand, then the expectation over $\varepsilon^{(t)}$ is equal to $b^2 \operatorname{Id}$.

Let $s(T)$ denote the number of good tasks in $T$, for any $T \subseteq \{1, 2, \dots, k\}$. Thus,

$$\mathbb{E}\left[\|u_T\|^2\right] = \frac{p\left(a^2 s(T) + b^2(\alpha - s(T))\right)}{\alpha^2 d}. \tag{37}$$

To argue about the deviation error of $\|u_T\|^2$, we use the following two estimates (see, e.g., Vershynin (2011)), which holds with high probability:

$$\left| (\varepsilon^{(j)})^\top X_j X_j^\top \varepsilon^{(j)} - \mathbb{E}\left[(\varepsilon^{(j)})^\top X_j X_j^\top \varepsilon^{(j)}\right] \right| \lesssim p\sqrt{d}a^2, \text{ for any } j = 1, \dots, k;$$

$$\left| (\varepsilon^{(i)})^\top X_i X_j^\top \varepsilon^{(j)} \right| \lesssim p\sqrt{d}a^2, \text{ for any } 1 \le i < j \le k.$$

Therefore, we get that for any $T \in \mathcal{U}$,

$$\left| \|u_T\|^2 - \mathbb{E}\left[\|u_T\|^2\right] \right| \le \frac{p\sqrt{d}a^2}{d^2}. \tag{38}$$

To finish the proof, consider a good task $i$ versus a bad task $j$. We need the gap in the expectation term between the good/bad tasks to dominate the standard deviation from the error terms. The gap in the expectations is based on equation (37). The standard deviation terms are upper bounded by the sum of equations (35) and (38).

Thus, provided that

$$(1 - 3\varepsilon_1)\frac{p(a^2 - b^2)}{\alpha^2 d} \ge (1 + 3\varepsilon_1)\frac{p\sqrt{d}a^2}{d^2} + \frac{2Ck\delta^{-1/2}}{\sqrt{n}}, \tag{39}$$

there must exist a threshold separating all the good tasks from the bad ones. We can verify that condition (39) is satisfied when

$$n \gtrsim C^2 \cdot k^2 \cdot \frac{1}{(a^2 - b^2)^2}, \quad \text{and}$$

$$d \gtrsim \left(\frac{a^2}{a^2 - b^2}\right)^2 k^4 + k \log\left(\frac{2k}{\delta}\right) + p.$$

To apply Algorithm 1, we set the threshold $\gamma$ as $k/\alpha$ times any value between the left-hand and right-hand side of equation (39) (recall that $k/\alpha$ is inherited from Lemma 3.3). Thus, when $n$ and $d$ satisfy the condition above, combined with Lemma 3.3, with high probability, for any $i$ such that $\hat{\theta}_i < \gamma$, $i$ must be a good task. When $\hat{\theta}_i > \gamma$, $i$ much be a bad task. Thus, we have finished the proof. $\qquad\square$

## C  Experiment Details

We describe details that were left out of the paper's main text. First, we describe the additional experimental setup and the implementation specifics. Second, we present results to further validate the sample complexity of task modeling. Third, we provide the experimental results that are omitted from Section 4, including the results for fairness measures and ablation studies.

### C.1  Implementation Details

For evaluating multitask learning with natural language processing tasks, we collect twenty-five tasks from several benchmarks, including GLUE, SuperGLUE, TweetEval, and ANLI. Due to the computation constraint, we did not include the tasks with a training set size larger than 100k. The collection spans numerous categories of tasks, including sentence classification, natural language inference, and question answering. Table 4 shows the statistics of the twenty-five tasks.

Table 4: Dataset description and statistics of twenty-five text datasets.

| Task | Benchmark | Train. Set | Dev. Set | Task Category | Metrics |
|---|---|---|---|---|---|
| CoLA | GLUE | 8.5k | 1k | Grammar acceptability | Matthews corr. |
| MRPC | GLUE | 3.7k | 1.7k | Sentence Paraphrase | Acc./F1 |
| RTE | GLUE | 2.5k | 3k | Natural language inference | Acc. |
| SST-2 | GLUE | 67k | 1.8k | Sentence classification | Acc. |
| STS-B | GLUE | 7k | 1.4k | Sentence similarity | Pearson/Spearman corr. |
| WNLI | GLUE | 634 | 146 | Natural language inference | Acc. |
| BoolQ | SuperGLUE | 9.4k | 3.3k | Question answering | Acc. |
| CB | SuperGLUE | 250 | 57 | Natural language inference | Acc./F1 |
| COPA | SuperGLUE | 400 | 100 | Question answering | Acc. |
| MultiRC | SuperGLUE | 5.1k | 953 | Question answering | $F1_a$/EM |
| WiC | SuperGLUE | 6k | 638 | Word sense disambiguation | Acc. |
| WSC | SuperGLUE | 554 | 104 | Coreference resolution | Acc. |
| Emoji | TweetEval | 45k | 5k | Sentence classification | Macro-averaged F1 |
| Emotion | TweetEval | 3.2k | 374 | Sentence classification | Macro-averaged F1 |
| Hate | TweetEval | 9k | 1k | Sentence classification | Macro-averaged F1 |
| Irony | TweetEval | 2.9k | 955 | Sentence classification | $F1^{(i)}$ |
| Offensive | TweetEval | 12k | 1.3k | Sentence classification | Macro-averaged F1 |
| Sentiment | TweetEval | 45k | 2k | Sentence classification | Macro-averaged Recall |
| Stance (Abortion) | TweetEval | 587 | 66 | Sentence classification | Avg. of $F1^{(a)}$ and $F1^{(f)}$ |
| Stance (Atheism) | TweetEval | 461 | 52 | Sentence classification | Avg. of $F1^{(a)}$ and $F1^{(f)}$ |
| Stance (Climate) | TweetEval | 355 | 40 | Sentence classification | Avg. of $F1^{(a)}$ and $F1^{(f)}$ |
| Stance (Feminism) | TweetEval | 597 | 67 | Sentence classification | Avg. of $F1^{(a)}$ and $F1^{(f)}$ |
| Stance (H. Clinton) | TweetEval | 620 | 69 | Sentence classification | Avg. of $F1^{(a)}$ and $F1^{(f)}$ |
| ANLI (A1) | ANLI | 1.7k | 1k | Natural language inference | Acc. |
| ANLI (A2) | ANLI | 4.5k | 1k | Natural language inference | Acc. |

Next, we report the results for baselines by running the open-sourced implementations from the respective publications. We describe the hyperparameters for baselines as follows. For higher-order approximation and task affinity grouping, we compute the task affinity scores between source and target tasks. Then, we select $m$ tasks with the largest task affinity scores as source tasks for each target task. $m$ is searched between 0 and the number of total tasks.

For gradient decomposition, we search the number of decomposition basis and auxiliary task gradient direction parameters, following the search space in Dery et al. (2021).

For weighted training, we search the task weight learning rate in $[10^{-2}, 10^2]$. The hyper-parameters are tuned on the validation dataset by grid search. For each target task, we search 10 times over the hyper-parameter space. We use the same number of trials in tuning hyper-parameters for baselines.

Table 5: Accuracy/Correlation scores on the development set using surrogate modeling followed by thresholding (ours), as compared with STL and MTL methods.

| Dataset
Metrics | CoLA
Matthews Corr. | RTE
Accuracy | CB
Accuracy | COPA
Accuracy | WSC
Accuracy |
|---|---|---|---|---|---|
| Train | 8500 | 2500 | 250 | 400 | 554 |
| Validation | 1000 | 3000 | 57 | 100 | 104 |
| STL | 59.38±0.70 | 67.94±0.74 | 70.36±1.82 | 64.00±2.19 | 60.00±2.76 |
| Naive MTL | 57.11±0.81 | 69.31±0.97 | 71.78±1.39 | 66.00±2.02 | 58.20±1.98 |
| HOA | 60.09±0.75 | 69.03±2.03 | 80.71±2.62 | 67.20±2.56 | 61.35±3.12 |
| **Alg. 1 (Ours)** | **60.43±0.79** | **70.83±1.97** | **83.57±2.43** | **69.20±3.71** | **65.38±2.31** |

Table 6: Violation of two fairness-related measures (demographic parity and equality of opportunity) on six multi-group learning tasks with tabular features, averaged over ten random seeds. **Lower is better.**

| Demographic parity | HI | KS | LA | NJ | NV | SC |
|---|---|---|---|---|---|---|
| STL | 12.95±1.76 | 4.09±1.15 | 26.30±1.21 | 26.06±0.53 | 12.62±1.99 | 22.51±0.47 |
| Naive MTL | 8.25±1.31 | 4.06±1.17 | 21.24±0.66 | 27.73±0.94 | 13.35±0.51 | 18.83±0.80 |
| HOA | 8.63±2.95 | 6.15±3.00 | 22.83±0.53 | 26.14±0.29 | 13.15±0.64 | 19.39±1.05 |
| TAG | 8.93±2.35 | 3.97±0.61 | 20.72±0.86 | 25.21±0.68 | 12.24±0.82 | 18.77±0.85 |
| TAWT | 18.12±1.80 | 4.84±0.71 | 25.77±0.94 | 25.66±0.38 | 12.40±0.74 | 23.16±0.42 |
| **Alg. 1 (Ours)** | **7.63±2.12** | **1.06±0.62** | **17.25±1.13** | **24.96±0.63** | **11.34±1.31** | **17.66±0.80** |
| Equality of opportunity | HI | KS | LA | NJ | NV | SC |
| STL | 9.86±1.29 | 1.43±3.62 | 29.64±3.24 | 22.43±1.02 | 13.61±3.67 | 29.93±0.77 |
| Naive MTL | 3.86±0.84 | 2.03±2.11 | 21.26±1.35 | 24.43±1.49 | 12.14±2.21 | 21.22±1.75 |
| HOA | 3.55±2.85 | 4.34±3.18 | 22.88±1.72 | 22.98±1.18 | 12.92±2.23 | 23.31±1.77 |
| TAG | 4.27±0.25 | 1.18±0.97 | 20.66±1.43 | 21.89±0.69 | 11.66±1.58 | 19.89±1.10 |
| TAWT | 4.21±2.25 | 1.40±2.14 | 30.38±2.17 | 23.26±0.30 | 11.77±1.01 | 30.86±0.84 |
| **Alg. 1 (Ours)** | **0.24±1.32** | **0.21±1.34** | **14.14±2.32** | **21.48±0.90** | **9.65±3.49** | **18.54±1.61** |

## C.2 Omitted Results from Section 4.3

**Complete results for NLP tasks.** In Table 5, we report the complete experimental results for applying our approach to NLP tasks, as reported in Section 4.3.

**Optimizing fairness-related metrics.** We show that task modeling is applicable to various performance metrics for capturing task affinity. Besides the average performance and worst-group performance discussed in Section 4.3, we consider two fairness measures: demographic parity and equality of opportunity (Ding et al., 2021).

The demographic parity measure is defined as:

$$\left| \Pr\left[\hat{y} = 1 \mid g = \text{black}\right] - \Pr\left[\hat{y} = 1 \mid g = \text{white}\right] \right|,$$

which measures the difference in the positive rates between white and African American demographic groups.

The equality of opportunity measure is defined as:

$$\left| \Pr\left[\hat{y} = 1 \mid y = 1, g = \text{black}\right] - \Pr\left[\hat{y} = 1 \mid y = 1, g = \text{white}\right] \right|,$$

which measures the difference in the true positive rates between the two groups.

We consider the binary classification tasks with multiple subpopulation groups. Table 6 shows the comparative results. First, similar to the worst-group accuracy results, we find that multitask approaches (including ours and previous methods) decrease the violation of both fairness measures compared to ERM, suggesting the benefit of combining related datasets. Second, our approach consistently reduces both fairness measure violations more by **1.26%** and **2.31%** on average than previous multitask learning approaches, respectively.

## C.3 Comparing the Running Time

We provide a comparison of the running time between our approach and the baselines in Table 7. We notice that the running time of our approach is comparable to MTL optimization methods after adding early stopping and downsampling to reduce the training time. Our approach is slightly slower than weak supervision methods that directly aggregate the weak labels while achieving 5% better performance on average.

| Dataset (Metrics) | CDR (Hours / F1) | Chemprot (Hours / Acc.) |
|---|---|---|
| Naive MTL | 1.99 / 58.20±0.55 | 1.89 / 53.43±0.53 |
| Majority voting | 2.00 / 58.89±0.50 | 1.91 / 57.32±0.98 |
| Probabilistic modeling | 2.00 / 58.48±0.73 | 1.91 / 57.00±1.20 |
| MetaL | 2.00 / 58.48±0.90 | 1.91 / 56.17±0.66 |
| TAWT | 2.30 / 59.85±0.30 | 2.02 / 53.76±2.96 |
| Auto-$\lambda$ | 3.46 / 59.07±0.05 | 3.31 / 52.50±1.28 |
| Alg. 1 w/o early stopping and downsampling | 38.34 / 61.22±0.39 | 31.14 / 57.54±0.55 |
| Alg. 1 w/ early stopping and downsampling | 2.89 / 60.77±0.05 | 3.76 / 57.06±0.84 |

Table 7: Running time of our approach after adding early stopping and downsampling to speed up the computation of $f(S_1), f(S_2), \ldots, f(S_n)$, as compared with baseline approaches.

## C.4 Ablation Studies for Constructing Surrogate Models

**Loss function:** We consider three choices of prediction losses, including zero-one accuracy, cross-entropy loss, and classification margin. We observe that the classification margin is more effective than the other two metrics. The Spearman's correlation of using the margin is 0.86 on average over two tasks (HI and LA). In contrast, the Spearman's correlations of using the loss and accuracy are 0.61 and 0.34, respectively. Besides, we compare the task selection using the three metrics in Table 8. We find that using the margin outperforms the other two by 0.37% on average over the six target tasks in terms of worst-group accuracy.

| | HI | KS | LA | NJ | NV | SC |
|---|---|---|---|---|---|---|
| $f$ uses zero-one accuracy | 75.16±0.70 | 76.39±1.09 | 75.15±0.43 | 77.40±0.49 | 74.34±1.81 | 77.29±0.19 |
| $f$ uses cross-entropy loss | 75.33±0.80 | 75.82±0.60 | 74.19±1.37 | 77.51±0.35 | 74.55±1.60 | 77.21±0.27 |
| $f$ uses classification margin | 75.47±0.73 | 76.96±0.69 | 75.62±0.11 | 78.17±0.36 | 75.21±0.52 | 77.62±0.34 |

Table 8: Choosing different loss functions $\ell$ for six target tasks in the multi-group learning setting.

**Subset size:** Recall that we collect training results by sampling $n$ subsets from a uniform distribution over subsets of a constant size. We evaluate the MSE of task models by varying $\alpha \in \{2, 5, 10, 20\}$. To control the computation budget the same, we scale the number of subsets $n$ according to $\alpha$. We train $n = 800, 400, 200, 100$ models with $\alpha = 2, 5, 10, 20$, respectively. We observe similar convergence results as in Figure 7. Among them, $\alpha = 5$ yields a highest Spearman's correlation of 0.89 between $f(\cdot)$ and $g(\cdot)$.

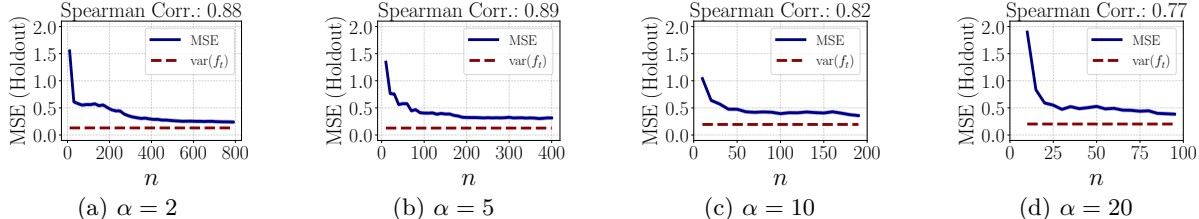

Figure 7: Fitting surrogate models using different $\alpha$ evaluated on a fixed target task.

**Number of sampled subsets:** Lastly, we show that task selection remains stable under different values of $n$. We measure the effect on two tasks (HI and LA) by comparing the 10 tasks with the smallest coefficients estimated from $n = 100, 200, 400$ subsets. We observe that using 100 subsets identifies 7/10 source tasks compared with $n = 400$. Increasing $n$ to 200 further identifies 9/10 source tasks compared with $n = 400$.

