# OpenReview forum: "Identification of Negative Transfers in Multitask Learning Using Surrogate Models"
_TMLR — Accepted by TMLR_

### Review · Reviewer_ZoXn · 2023-01-11

**Summary Of Contributions:**

This paper aimed to improve the performance of multi-task learning problems by identifying the valuable subsets for the target task due to the wide existence of negative transfers. It proposed a simple algorithm that mainly includes two key steps: 1) sample some randoms subsets from source tasks and leverage them to train multiple surrogate models, respectively; 2) use the scores predicted by the surrogate model for each task as the standard to select the beneficial subsets to train the final model for the target task. Finally, theoretical analysis and comprehensive experiments are conducted to show the effectiveness of the proposed method.

**Audience:**

Yes

**Broader Impact Concerns:**

I don't have such concerns

**Claims And Evidence:**

Yes

**Requested Changes:**

1.	In Fig.1 and Alg.1, what is the surrogate model? What are its parameters? In the descriptions of the paper, it seems that the surrogate model $g_{\theta}$ is just combined by some scalar value $\theta_i$ for all k tasks. How do we generalize the proxy $g$ to the unseen task selections?
2.	In Sec.2.2, it is unclear whether we need to train the surrogate model by sampling n subsets and then use the model to select the positive sources tasks to train the final prediction model for the target or both of the models are trained simultaneously. Thus, some of the method sections should be reorganized.
3.	In Eq.(5), why the expectation of Rademacher complexity is taken by $\tilde{D}$ instead $D_t$?
4.	In Thm.2.1, what is the difference between $\alpha$ and $m$? And how is the probability of at least 0.99 obtained?
5.	Each subset $S_i$ with size $\alpha$ would be independently used to train a surrogate model. But I notice that the practical $\alpha$ is relatively small, which seems insufficient to train a deep neural network model. How to deal with this case in the paper?
6.	In my opinion, sampling multiple random subsets to train surrogate models is time-consuming and might be unaffordable in the practical deployment of deep learning models. The authors should provide the practical training time for both the proposed algorithm and the baseline models to show the efficiency.
7.	What is the effect of $\gamma$ to the performance of negative transfers?
8.	Some new SOTA competitors that suppress the negative transfers in MTL should be compared, such as [1, 2].
9.	This paper attempts to address the negative transfer problem across the task perspective, but the negative transfer across features is also essential. Is the proposed method available for this?


**Strengths And Weaknesses:**

The motivation of this paper is good. This paper considered a critical problem in a multi-task learning problem, i.e., negative transfer, which has recently received much attention in the machine learning community.
The proposed algorithm seems promising, supported by rigorous theoretical guarantees and empirical studies.

However, there are still some weaknesses in the work:
1.	The contribution of this work is not clear. The authors should give more explanations about this, particularly the strengths of the proposed method against the existing literature targeting negative transfers.
2.	Some parts of the paper are difficult to understand and need to be improved. Details of them will be listed in the next part.
3.	The related work for solving negative transfers in MTL is incomplete. For example, another essential line to address negative transfers in MTL is called block-diagonal structural learning, which aims at promoting positive information across all tasks and suppressing negative knowledge sharing in the learning process [1,2]. I think the authors should consider this and discuss the strengths of the proposed method.
4.	The efficiency of the proposed algorithm is a significant concern, but it has not been well verified in the paper.

Ref:
[1] Task-Feature Collaborative Learning with Application to Personalized Attribute Prediction.
[2] Calibrated multi-task learning.

---

> ### Author Response · Authors · 2023-02-15
> **Response (Part I)**
>
> Thanks for providing detailed feedback. We first address the reviewer’s high-level comments. Then, we describe our revision in response to the requested changes.
>
> **Contribution of our work and strengths of the proposed method.** As the reviewer has pointed out, our work considers the problem of negative transfer, which is a critical issue in multi-task learning. A naive approach to address this problem is by enumerating all possible subsets of source tasks, which amounts to training a total of $2^k$ models. Clearly, this is very expensive, e.g., most of the datasets in our experiments have more than $k \ge 20$ source tasks. Therefore, to better approach this problem, several prior works such as Standley et al.’20 and Fifty et al.’21 have studied the relationships between tasks to provide insights. Both of these works rely on the computation of *first-order task affinity scores*, which measure the pairwise transfer relationship from one task to another task.
>
> Our paper continues this line of work, and our first contribution is to model higher-order task relationships, which is the transfer relationship from a set of tasks to another task. Concretely, we model this with a linear regression model, parametrized by a set of relevance scores, one for each task. Interestingly, this simple approach remains consistently accurate for predicting positive/negative transfers from (up to) twenty source tasks to one target task, as shown in Figure 3. By contrast, this is not captured by pairwise transfer relationships alone.
>
> Our second contribution is to design a subset selection algorithm for multi-task learning, particularly for optimizing the performance of a given target task. Compared with the existing methods, we believe that the strength of our approach is that our approach can identify positive/negative transfers much more accurately. As an additional comment, we think that our approach is quite different from both references mentioned by the reviewer; Our approach can be viewed as a simple instantiation of “meta-learning” the multi-task transfer relationships.
>
> Our third contribution is to show extensive theoretical and empirical results to justify our approach. To highlight these aspects, we have added a paragraph at the end of the introduction to summarize the contribution of our paper. We apologize if this does not come across as clearly in the initial submission.
>
>
>
> **Clarification of the surrogate model and various aspects of our approach**. We again apologize for the confusion in the initial submission. Our notion of surrogate models follows a recent paper on datamodels by Ilyas et al. (ICML (2022)), and our design of a linear regression model is also inspired by that paper. The parameters of the surrogate model are simply $\theta_1, \theta_2, \dots, \theta_k$. To apply the proxy to an unseen subset, say $S$, we will sum up the scores: $g_{\theta}(S) = \sum_{i \in S} \theta_i\$.
>
> Our subset selection method first trains the surrogate model; This is described in Section 2. After training the surrogate model, we then use the relevance scores learned by training this model to perform subset selection. This is described in Section 3.
>
>
>
> **Additional related works.** Thanks for suggesting the related works.
>
> First, we note that reference [1] is based on applying an explicit regularization on the feature space of all the tasks. We have cited several earlier papers that use this idea, including Evgeniou and Pontil (2004), Argyriou et al. (2007,2008), and Kumar and Daume (2012). We have now also added [1] to our references too.
>
> Second, while reference [2] also studies linear regression tasks, note that their approach requires each source task to follow a linear regression structure and is not directly applicable to deep neural networks. We instead use the linear regression model as a “meta-learning” model to extrapolate the task relationships but stress that our approach can be applied to non-linear tasks.
> We have cited both papers in the related works in the revised version.

---

> ### Author Response · Authors · 2023-02-15
> **Response (Part II)**
>
> **Efficiency of our approach.** In the initial version, we reported the runtime of our approach in Section 4.2. We have revised that section to make the comparison clear. We show that the number of training hours for our approach is less than the two most related baselines. Concretely, the baseline takes more than 200 hours on datasets with more than 20 tasks due to the search over the exponential task space. By contrast, our approach takes at most 49 hours.
>
> In our experience, the linear surrogate model has been sufficiently efficient for us to train. Recognizing the reviewer’s concern regarding efficiency, we have further explored ways to speed up our approach using two techniques.
>
> - The first technique is to train each network for fewer iterations (i.e., early stopping).
> - The second technique reduces the amount of training data for fitting each MTL model by downsampling the combined training data of all tasks.
>
> Below, we show that we can achieve 12x speedup after adding both techniques while achieving comparable test performances. These results have now been included in Section 4.2.
>
> | Runtime for training a surrogate model (by hours)   | Chemprot |  CDR  |
> | --------------------------------------------------- | :------: | :---: |
> | Our approach	w/o early stopping and downsampling |  38.34   | 31.14 |
> | Our approach w/ early stopping and downsampling     |   2.89   | 3.76  |

---

> ### Author Response · Authors · 2023-02-15
> **Response (Part III)**
>
>
>
> **Regarding the requested changes:**
>
>
>
> **Better explaining the surrogate model (1,2).** In the revised paper, we have clarified various sentences in the caption of Figure 1 and the introduction to clarify the definition of the surrogate model. We have included another paragraph in Section 2.1 to give some context about linear surrogate models.
> One surprising finding from our work is that we can predict positive/negative transfers for unseen task combinations by fitting a linear surrogate model. We have noted an F1-score of at least 0.8 (see Figure 3).
>
>
>
> **Questions regarding our theoretical results (3,4)**.
>
> - Recall that $\tilde{D}_t$ refers to the empirically-drawn training data samples. When we define the Rademacher complexity of a function class, the expectation should be taken over the randomness of the training data samples. We have added two sentences to clarify this definition.
> - $\alpha$ is the size of each subset $S$. $m$ is the size of the validation dataset used to evaluate an MTL model’s validation loss on the target task. The probability value of $0.99$ is by setting the value of $\delta$ to be less than $0.005$ in both Lemma 2.2 and Lemma 2.3. More generally, our result can be applied to different levels of confidence values. We have added one sentence to clarify this point.
>
>
>
> **Questions regarding the size of $\alpha$ (5).** For each task subset with size $\alpha$, we combine the training data from $\alpha$ tasks and train a model on the combined dataset. On the datasets in our experiments, the training size of combined $\alpha$ tasks is at least $200$ samples, which is sufficient to train the model. We have added a sentence to clarify this in the revised manuscript.
>
>
>
> **Runtime (6)**. Thanks for the comment, We have included a runtime comparison with baselines and incorporated two techniques (namely, early stopping and downsampling) to reduce training time in Section 4.2 of the revised manuscript.
>
>
>
> **Effect of $\gamma$ (7).**  The parameter $\gamma$ is only used for selecting tasks to train with the primary target task in the final prediction model. A larger $\gamma$ selects more source tasks since we select source tasks whose relevance scores are smaller than $\gamma$. The $\gamma$ is tuned between -0.5 and 0.5; this range covers most values corresponding to the relevance scores.
>
> We observe that for text datasets with noisy labels, the threshold is set close to 0.5, which selects most source tasks from the data. For NLP and multi-group fairness tasks, the threshold is set close to -0.5, which selects less than half of the source tasks. We have added one sentence to describe the effect of $\gamma$ in Section 4.5.
>
>
>
> **Comparison with references [1,2] (8).** Thanks for suggesting the references. We have included both papers in the related works in the revised submission.
>
>
>
> **Feature selection (9)**. Thanks for the great suggestion. It is possible to apply our approach to feature selection. In particular, one can set up a method that samples several features each time to fit one model and then aggregate the trained models from multiple sampling steps. This would be an interesting research question for future work.

---

### Review · Reviewer_HbzV · 2023-01-27

**Summary Of Contributions:**

This work explores task selection in multi-task learning: given a target objective, which subset of a collection of source tasks would most improve the performance of the target?

This is a very challenging problem; non-linear interactions among different tasks often makes it difficult to identify such a collection of source tasks. For instance, while tasks A and B may individually help task X; training X together with A and B may actually result in performance degradation than from training X by itself.

To this end, the authors propose a three-step algorithm:
(1) randomly sample a set of task sequences from the collection of source tasks. Let's say $n$ tasks sequences are sampled in total.
(2) For each of the $n$ task sequences, fully train a multi-task model where the targets are the objectives in this sequence *and* the target objective (i.e. the task whose performance we would like to maximize).
(3) Train a surrogate model to predict the influence of this sequence of source tasks on the target task.

During test time, one can limit the sequence of source tasks to a single task to measure if this task offers "positive" or "negative" transfer on the target task.

**Audience:**

Yes

**Claims And Evidence:**

Yes

**Requested Changes:**

Please simplify the writing. Consider saying less to communicate more. Make it super simple, and clean up the notation to better convey your meaning. I'm also guilty of falling into this trap myself -- when you invest significant time into a project and think about it all the time -- it can be difficult to recognize another person/reader does not have the same context you do. The best & strongest works try to convey the readers understanding & familiarity into the minds of their audience.

I'm currently marking "No" to both "Claims and Evidence" as well as "Audience" sections, as I do not understand your approach, and cannot confidently answer these questions in the affirmative. I suspect other readers may experience similar confusion.

You may also consider citing the following related work. While I don't think it's necessary to compare against its approach, a sentence in the related work about the (dis)similarity to your own approach would be appropriate:

[1] Auto-λ: Disentangling Dynamic Task Relationships, Liu et al.

------ Updated Response 2/16/23 -----

Given the author's elucidating response and changes to their manuscript, I strongly endorse the publication of this work.

**Strengths And Weaknesses:**

From the length of this paper, the mathematical analysis, and experimental results sections, it is clear the authors have invested significant time and effort into developing and evaluating their approach. I applaud their presentation of the experimental findings; it seems clear that their method "works".

I have one suggestion for improvement: **make the manuscript more simple.**

While I consider myself familiar with this space, it was surprisingly difficult for me to understand the proposed method, approach, and motivation. There seems to be a lack of structure to the abstract, introduction, and other sections of this paper which makes it difficult for a reader unfamiliar with this work to understand the message it seeks to convey.

For instance, you might simplify your abstract like:

(1) [Multi-task is useful for objectives where labeled data is limited]:
Multi-task learning is widely used throughout industry and academia to fit a training objective where labeled data is limited by training this objective alongside other objectives where data is more abundant.

(2) [But identifying these other objectives is difficult]:
However, identifying the training objectives that would improve the performance of the task with limited labeled data is difficult.

(3) [It is difficult because source tasks can interfere with the training of the target task.]:
...

(4) [We propose a solution that models the association between a subset of source tasks and a target task by leveraging a surrogate model]:
...

(5) [We find this approach to predict negative transfer between source tasks and the target task]:
...

I also struggle to understand the formulation of the surrogate model.

Figure 1 describes selecting a subset of task groupings, fully training a multi-task learning model with these groupings and the target task to compute $f(S_i)$: the performance of the target task when trained together with the tasks in $S_i$, and then a surrogacy model is introduced? What is this surrogate model? Why are you using $\theta$ to indicate relevancy scores rather than its traditional usage as a model's parameters? What is $g_\theta(S_i)$? What are the surrogacy model's inputs and outputs, and ow is it paramterized (i.e. what are its parameters -- a shallow MLP, CNN, etc.)? Later in the page, it says $g_\theta(S) = \sum_{j\in S} \theta_j$, where $\theta_j$ are the relevancy scores.

But in one paragraph above, "To estimate the relevance scores, we introduce a surrogate model $g_\theta(S)$,  parametrized by the relevance scores, to approximate MTL performances." To me, this means you parameterize the model by the relevancy scores (i.e. these are the model's parameters), and you're using this model to predict the relevancy scores themselves?

---

> ### Author Response · Authors · 2023-02-16
> **Response (Part I)**
>
> We thank the reviewer for carefully reading our paper and providing detailed suggestions! We realize the reviewer’s concern regarding the exposition of this work, so we have made significant revisions to the paper by following the reviewer’s suggestions, which are marked in pink color in the updated version. Below, we will explain the high-level idea of our approach and compare our approach with the approach from the reference [1]. We hope these discussions will help the reviewer better understand our work.
>
> **Better explaining the surrogate model.**  Our notion of surrogate models is inspired by a recent paper about datamodels (Ilyas et al. (2022)). To give some context, in that paper, the authors train a linear regression model to predict the outcome of a deep neural network trained on a subset of the training data. An empirical finding from that paper is that this linear regression model is a good fit on several popular benchmark datasets, including CIFAR.
>
> Our design of surrogate models is similar to that paper, but we extend this idea to multi-task learning. To set up this model, suppose we choose a subset of source tasks, say $S$, then the covariates for this subset is a zero-one vector, and the $i$-th coordinate is $1$ if $i \in S$, and is $0$ otherwise. The label for this subset $S$ would be $f(S)$, which is the MTL performance of combining all the data of tasks in $S$ with the target task, evaluated on the target task. These are the feature covariates and the label of one subset $S$.
>
> We use a linear regression method to approximate $f(S)$. This linear regression model has $k$ parameters: $\theta_1, \theta_2, \dots, \theta_k$, each representing the relevance score of one source task to the target task. These relevance scores are the only parameters in the linear model $g_\theta(S)$.
>
> To learn these $k$ parameters, we precompute the value of $f(S)$ for $n$ random subsets. In practice, $n$ is somewhere between $100$ to $200$. We can use early stopping and downsampling to speed up the computation of each $f(S)$. Then, we minimize the mean squared error between $g_{\theta}(S) = \sum_{i \in S} \theta_i$ and $f(S)$, averaged over $n$ random subsets.
>
> After learning these $k$ parameters, we predict an unseen subset $S$'s MTL performance as $g_{\theta}(S)$, using the fitted parameters. In Figure 6 from Appendix A, we validate this linear regression model across 16 different datasets on a holdout set of 100 random subsets; We find that the Spearman correlation between linear model predictions and the true MTL performances is above 0.8, averaged over all 16 datasets.
>
> Our approach is conceptually similar to the construction of random forests. In the first step, we randomly sample subsets of source tasks to train MTL models together with the target task. In the second step, we aggregate these results using the linear regression model. This aggregation step is similar to averaging the MTL performances of all subsets that include a particular task. This interpretation is derived from our theoretical analysis in Lemma 3.3.
>
> We apologize if this does not come across as clearly in the initial submission and are sorry for any confusion. We have revised the manuscript by adding explanations and expanding various descriptions accordingly.
>
> **Writing changes**. We are very grateful for the reviewer’s careful reading of our manuscript. Below is a list of revisions we have added to the paper.
>
> - In the abstract, we have made several revisions and expanded on the aspect of identifying negative transfers from a set of source tasks to another target task.
> - In the introduction, we have added descriptions to clarify the notion of surrogate models.
> - In the main text, we have made several revisions to better highlight the context as well as our approach.

---

> ### Author Response · Authors · 2023-02-16
> **Response (Part II)**
>
> **Comparison with related work [1].** Thanks for suggesting this related work. Based on our understanding, this paper designs a bilevel optimization method to optimize model parameters and task-specific weights jointly. We think our approach can be complementary to this approach because our approach is focused on subset selection, whereas this paper focuses on reweighting the source tasks. For example, one could imagine combining both approaches by first applying subset selection, and then reweighting the selected tasks.
>
> One potential advantage of our approach would be in settings where some source tasks are very noisy. For example, in the weak supervision datasets, we have several weak labels for each unlabeled data point, each annotated based on a labeling function. These labeling functions can be quite noisy, depending on the dataset. In the table below, we compare our approach with the approach from reference [1] (Auto-$\lambda$) on five weak supervision datasets; See Section 4.1 for the description of these datasets.
>
> | Dataset (Metric)      | Youtube (Acc.) |  TREC (Acc.)   |    CDR (F1)    | Chemprot (Acc.) | Semeval (Acc.) |
> | --------------------- | :------------: | :------------: | :------------: | :-------------: | :------------: |
> | \# Labeling functions |       10       |       68       |       33       |       26        |      164       |
> | Auto-$\lambda$        | 95.80$\pm$0.85 | 73.70$\pm$0.67 | 59.07$\pm$0.05 | 52.50$\pm$1.28  | 87.91$\pm$0.66 |
> | Our approach          | 97.47$\pm$0.82 | 81.80$\pm$1.14 | 61.22$\pm$0.39 | 57.54$\pm$0.55  | 93.50$\pm$0.24 |
>
> We observe that for these weak supervision datasets, which involve a lot of noisy labeling functions, our approach generally achieves greater performance. In Figure 5 from Section 4.3, we can further see that our approach indeed selects labeling functions that are more accurate. We have now included these results and a discussion of this work in the revised submission.
>
> **Response to other comments**
>
> **\>>> "During test time, one can limit the sequence of source tasks to a single task to measure if this task offers "positive" or "negative" transfer on the target task."**
>
> We would like to clarify that our approach can also measure the transfer from a set of source tasks to another task. This is done by summing up the relevance scores of each source task in the set. In Figure 3, we show that this approach can be used to predict positive/negative transfers from one set of tasks to another task, with up to 80% $F_1$-score.
>
>
>
> **\>>> "There seems to be a lack of structure to the abstract, introduction, and other sections of this paper which makes it difficult for a reader unfamiliar with this work to understand the message it seeks to convey."**
>
> There are three claims we would like to make in this paper.
>
> 1. The design (and analysis) of linear surrogate models; This is described in Section 2.
> 2. The subset selection algorithm and its analysis, described in Section 3.
> 3. The experiments for our approach: We first show our results for the surrogate models in Section 4.2. Then, we show the results for various MTL datasets in Section 4.3-4.4.
>
> Both the abstract and the introduction followed the logic of these three claims. We apologize if this logic did not come across as clearly in the initial submission, and we hope that this description helps clarify the structure of our paper.

---

> > ### Comment · Reviewer_HbzV · 2023-02-17
> > **Revised Review**
> >
> > Great response -- this clears up quite a bit of my confusion and the changes to the text are quite good. Accordingly, I revise my review to strongly endorse this work for publication.
> >
> > Coincidentally, a colleague who previously published on task grouping research reached out to me this past week to ask about adapting datamodels to predict task groupings. Their idea is similar to the research concept encapsulated in this work, and the findings and methods developed by this paper will be very helpful for their --- as well as other researchers in the field --- mental models and research questions going forward.

---

### Review · Reviewer_Nt3Y · 2023-02-05

**Summary Of Contributions:**

The paper considers the multitask learning setting wherein a set of "source" tasks/datasets are available to augment with when training on a "target" task/dataset.
Given a target task sample, the paper proposes a parameterized algorithm for identification/selection of an $\alpha$-sized subset of source tasks to add to the training set of the target sample.
The algorithm constructs a linear predictive model of transfer by minimizing squared loss on a dataset of size $n$ whose examples are 1-hot encodings of source task presence/absence and whose labels are model performance when trained on the selected sources + target datasets.
Then, the algorithm compares the linear coefficients of the predictive model, or source relevance scores, to a threshold $\gamma$ to select the source tasks for multitask training.

The accompanying guarantees show that the algorithm's estimated coefficients converge to the population coefficients and that a threshold value exists, under some conditions, for linear regression tasks. The experiments consist of three multitask learning settings: selecting labelers in a weak supervision setting, standard NLP multitask learning, and robust learning in a multi-group setting. The proposed algorithm shows improvement over single task, naive multitask, and other competitor baselines in all settings. Some ablations and further analysis suggest the proposed algorithm is not sensitive to its parameters, validates some settings selected for the experiments, and suggests the algorithm is somehow capturing the effect of higher-order interactions between source and target tasks.

**Audience:**

Yes

**Claims And Evidence:**

Yes

**Requested Changes:**

* What are the relative runtime differences b/w the proposed method and competitors listed in Table 1? Please include a summary of relative computational costs of the proposed method vs. competitors listed in the paper.

* Does the result of Theorem 2.1 only hold when one knows the target task is composed of exactly $\alpha$ source tasks?

* Based on their experiments, can the authors say
  * How often a set of source tasks not in the length-$n$ training set was selected?
  * How often a set of source tasks of size less than size $\alpha$ was selected? Greater than size $\alpha$?

* Theorem 3.1 describes the conditions under which a threshold exists. This justifies the final selection step of the proposed algorithm. How does the the superlinear growth in $m$ (w.r.t. $p$) compare to conditions required for computing a corresponding "good" linear predictor when performing single task learning on the target?

**Strengths And Weaknesses:**

* Overall, I thought this paper did a very good job in justifying the proposed algorithm. The theoretical evidence may not be very satisfying, but its development added to my understanding of the algorithm, and I appreciated the clarity of its exposition (both in the main and appendix). The experiments provided more convincing evidence considering relevant settings and showing consistent improvements across different multitask settings.
  * Given the proof approach, the Rademacher-based bound can't say much when the target task's validation sample is small, i.e., approaching the few-shot setting. However, looking at dataset sizes in the main tables and appendix, the proposed method does perform well at somewhat low validation set sizes. It might be good for the paper to briefly highlight this dimension of the method's performance.

* The potential weakness here is in the paper not discussing the limitations of the proposed method directly.
  * The proposed method's computational cost relative to competitors: the runtime cost in section 4.2 is fine to include but not sufficient to address this weakness.
  * Does the theoretical development shed light on the $(n, m)$ regime where the proposed algorithm would be preferred to single task learning on the target? E.g., among a set of linear prediction tasks as described for Theorem 3.1?

---

> ### Author Response · Authors · 2023-02-18
> **Response (Part I)**
>
> We are grateful to the reviewer for providing a detailed summary of our work and asking various insightful questions. Here is our response to the reviewer's comments and requested changes. Based on them, we have also made significant revisions since the initial submission. These revisions are marked in pink color in the revised submission. We hope that we have answered all the questions from the reviewer and are happy to discuss any other questions about our work.
>
> **About validation set size in the Rademacher-based bound.** Thanks for the great suggestion. The reviewer is correct that if the Rademacher complexity-based bound is too large, then when $m$ is very small, as in few-shot learning, our bound cannot say much about the convergence. To remedy this gap, we suspect that with some notion of data-dependent capacity bounds, such as the margin, we can get a tighter complexity bound for small scales of $m$. This would be an interesting question for future work.
>
> That being said, the main message we hope to convey from our Rademacher complexity-based bound is to show that constructing linear surrogate models has a linear asymptotic runtime complexity as a function of $k$, the number of source tasks. Concretely, we could expect our approach to scaling to very large numbers of source tasks; The largest in our experiment is 164, but we suspect that our approach might work for even larger numbers of source tasks, thanks to the linear scaling in $k$.
>
> We have added a sentence highlighting the question for the few-shot learning setting in the conclusion section.
>
> **Computational cost relative to the competitors.** The reviewer is correct to notice that since our approach requires training multiple MTL models (one for each subset), this would incur a computational overhead compared with naive MTL. In our experience, training these linear surrogate models has been efficient enough for our datasets. Concretely, on one dataset with as many as 164 source tasks, we have managed to use only 48 hours to train the surrogate model on a single GPU.
>
> We recognize the reviewer’s concern regarding efficiency, so we have explored two techniques to speed up our approach. These have now been added to Section 4.5 of the paper.
>
> - The first technique is to train each network for fewer iterations (i.e., early stopping).
>
> - The second technique reduces the amount of training data for fitting each MTL model by downsampling the combined training data of all tasks.
>
> Although these are simple/standard techniques, we show that they can be added to our approach to speed up the computation of $f(S_1), f(S_2), \dots, f(S_n)$. Below, we show a table that compares the number of training hours between our approach and baseline approaches. These are conducted on two weak supervision datasets, while similar results can be expected on the other datasets. These results have been added to Appendix C.3 of the revised manuscript.
>
> | Runtime (by hours) | Chemprot (Hours / Acc.) | CDR (Hours / F1) |
> | ------------------ | :---------------------: | :--------------: |
> | Naive MTL          |      1.89H / 53.43      |  1.99H / 58.20   |
> | MetaL              |      1.91H / 56.17      |  2.00H / 58.48   |
> | TAWT               |      2.02H / 53.76      |  2.30H / 58.85   |
> | Auto-$\lambda$     |      3.31H / 53.50      |  3.46H / 59.07   |
> | HOA                |     > 200H / 45.67      |  > 200H / 59.76  |
> | TAG                |     > 200H / 53.67      |  > 200H / 59.31  |
> | Our approach       |      2.89H / 57.06      |  3.76H / 60.77   |
>
> In terms of efficiency, our approach is now comparable to the optimization methods in the baselines while outperforming them in terms of accuracy/F1. Our approach is slightly slower than weak supervision methods that directly aggregate the weak labels into a strong label while achieving 5% better performance on average.

---

> ### Author Response · Authors · 2023-02-18
> **Response (Part II)**
>
> **How does our theoretical result compare with single task learning on the target task?** Thanks for the insightful question. Once we identify the related tasks from all tasks, we can combine them with the target task. This will yield better performance than single-task learning, provided the distance between the source-target tasks’ $\beta$-coefficients is small enough. In the revised manuscript, we have added a remark to emphasize this point in Section 3.
>
> **Other potential limitations of the proposed method**. This is a great question. A potential limitation of our method is that we are using a linear specification of the relevance scores in the surrogate model. We found that this is a reasonable model for the subset selection problem in our datasets since the goal is to make a binary decision of whether to choose a source task or not.
>
> If the goal is to cluster the tasks into multiple groups, or if the task relationships are very complex (again, this would depend on the specific dataset/application), then it might make sense to look at quadratic (or even cubic) interactions of the source tasks; See our footnote at the bottom of page 5. However, this would increase the time complexity of fitting the surrogate model to $k^2$. So there is a trade-off between the representation power of the model and the sample complexity for fitting it.
>
> **Regarding the requested changes**.
>
> **Comparing the running time.** We have added the following revisions to compare the computational cost of our approach with baselines.
>
> First, we have clarified the description in Section 4.2 by emphasizing that the runtime of our approach scales linearly to the number of tasks $k$. In contrast, the two most-related baselines have an exponential running time complexity. We have also revised Figure 4 to make this claim clear.
>
> Second, we have added a table to compare the cost of our approach and all the baselines conducted on two datasets in Appendix C.3.
>
> Third, we have added the result from using two speed-up techniques (namely, early stopping and downsampling) in Section 4.5. These techniques can speed up our approach by 12x while getting comparable performances.
>
> **Does the result of Theorem 2.1 only hold when one knows the target task is composed of exactly $\alpha$ source tasks?** Thanks for the excellent question. Our current proof does rely on the design of the $\alpha$-sized subsets. In particular, because the covariates of each subset are a zero-one vector, the population covariance of the entire space of $\alpha$-sized subsets becomes an identity matrix plus a rank-one matrix. See equation (12) in Section 3. Thus, the inverse of the covariance matrix (in the ordinary least squares estimator) is an identity matrix plus a rank-one matrix. We find that this design of random subsets has nice covariance structures, and we have added a remark to clarify this point in Section 2.
>
> **Questions regarding the selected tasks.** Our approach selects source tasks by choosing the tasks whose relevance scores are below some threshold. In practice, we find that the selected number of source tasks varies, and the selected set may also differ from the $n$ random subsets. Because of this, we also vary $\alpha$ for each dataset. We pick $\alpha$ between $\{3, 5, 10, 15\}$ via cross-validation, on a holdout set of $100$ subsets. We pick $n$ in $\{50, 200, 400, 800\}$ according to the number of tasks $k$.
>
> To give some examples, in the weak supervision datasets, we find that the number of selected source tasks is generally larger than $\alpha$. This is because most of the labeling sources turn out to be helpful for the target task. For example, on one dataset with 164 source tasks, our approach selected 160, while $\alpha$ is 15.
>
> For another example, in the NLP and multi-group fairness datasets, we find that the number of selected source tasks is generally less than $\alpha$. This indicates that the number of helpful source tasks is limited for these datasets. $\alpha$ is set as five on these datasets, while the selected tasks are usually 3-4 by the end.
>
> We have revised Section 4.6 to clarify this point in the manuscript.
>
> **Dependence on $m$ compared to conditions required for single-task learning.** Thanks for the insightful question. In terms of dependence on $m$, our result needs $m$ to be at least $p \log p$. This ensures that the concentration error of computing $f$ on the validation set becomes vanishingly small. If we were to apply single-task learning, we would need the scaling of $m$ to be at least $p \log p$ to ensure that the concentration error of $f$ on the validation set is small.

---

### Comment · Action_Editors · 2023-02-12
**Rebuttal reminder**

Dear authors,

This is a reminder that you will have two weeks' time to do rebuttals since three comments were submitted.

Best wishes,
AE

---

> ### Author Response · Authors · 2023-02-20
> **Summary of responses and revisions**
>
> Dear AE,
>
> Thanks for handling our paper. Based on the reviewer’s feedback, we have made the following revisions compared with our initial submission.
>
> **Better explaining the surrogate model:** We added a paragraph in Section 2 to explain the idea of surrogate modeling. Then, we clarified the contribution of our work in the abstract/introduction. We also expanded the description of our approach at various places in Section 2.
>
> **Clarifying the runtime comparison and improving the efficiency of our method:** We revised Section 4 to describe the runtime comparison and added another table in Appendix C. We also added techniques to speed up the training of surrogate models in Section 4.
>
> **Related work:** We added all the references suggested by the reviewers in Section 5. Besides, we included another comparison with recent work in Section 4. Lastly, we revised the related work discussion to improve the exposition in Section 5.
>
> **Other revisions:** Based on the reviewers’ detailed feedback, we added several remarks and explanations to clarify our theoretical results in Section 2/3. We expanded our discussion of various hyper-parameters in Section 4.
>
> We plan to release our code after the reviewing process for reproducing our results. We hope these responses and revisions help address the reviewers’ concerns, and we are happy to discuss any further issues.

---

### Decision · Action_Editors · 2023-03-14

**Recommendation:** Accept as is

**Comment:**

This paper presents work on multitask learning. It focuses on an important and challenging research problem in multitask learning: given a target objective, which subset of source tasks can most improve the performance of the target? To address the problem, the paper proposes a simple but effective idea that tends to determine which tasks may train well together in multi-task learning paradigms. Both theoretical and experimental results are provided to confirm the superiority of the proposed method.

Before rebuttal, reviewers raise several concerns about this paper, including the running time of the algorithm, supplementary explanations of surrogate models, and several questions on theory parts. The authors provide detailed and convincing responses to address the concerns. Reviewers are satisfied with the responses and consider that this paper is stronger after revisions. AC and all reviews appreciate the contributions of this paper. We therefore recommend an acceptance.

This paper is high quality and then selected for the Featured Certification, which is recommended by most reviewers. After the responses from the authors, the claims of this paper are clearly justified. AE and reviewers consider that this paper makes sufficient contributions to the research area of multitask learning, including its strong motivation, simple and effective idea, and meaningful theoretical results. This paper will largely inspire follow-up research. Therefore, it deserves the Featured Certification and should be highlighted.



**Audience:**

The paper will be of interest to researchers from the community of multitask learning.

**Claims And Evidence:**

The claims are supported by theory and empirical results.